# Identification of a binding site for small molecule inhibitors targeting human TRPM4

Babatunde Ekundayo [1], Prakash Arullampalam[2], Christian E. Gerber [2], Anne-Flore Hämmerli[2], Sabrina Guichard[2], Mey Boukenna[2], Daniela Ross-Kaschitza [2], Martin Lochner [2], Jean-Sebastien Rougier [2], Henning Stahlberg [1] ✉, Hugues Abriel [2] ✉ & Dongchun Ni [1,3]

Transient receptor potential (TRP) melastatin 4 (TRPM4) protein is a calcium-activated monovalent cation channel associated with various genetic and cardiovascular disorders. The anthranilic acid derivative NBA is a potent and specific TRPM4 inhibitor, but its binding site in TRPM4 has been unknown, although this information is crucial for drug development targeting TRPM4. We determine three cryo-EM structures of full-length human TRPM4 embedded in native lipid nanodiscs without inhibitor, bound to NBA, and an anthranilic acid derivative, IBA. We found that the small molecules NBA and IBA were bound in a pocket formed between the S3, S4, and TRP helices and the S4-S5 linker of TRPM4. Our structural data and results from patch clamp experiments enable validation of a binding site for small molecule inhibitors, paving the way for further drug development targeting TRPM4.

Transient receptor potential (TRP) ion channels, a superfamily of cation channels, play a crucial role in various physiological functions, including sensory perception, cellular homeostasis, and ion transport[1,2]. The potential of TRP channels as therapeutic targets for drug development is immense, given that mutations in the genes encoding these channels can lead to their dysfunction or dysregulation, implicating them in numerous diseases and genetic disorders[3,4]. This underscores the importance of our research in understanding the role of TRP ion channels, particularly TRPM4, and their potential as therapeutic targets for drug development.

The TRPM (melastatin-like transient receptor potential) subfamily member, TRPM4, is a calcium-activated non-selective monovalent cation channel. The voltage-dependent activation of TRPM4 by intracellular calcium drives a current due to monovalent cations flux such as $Na^+$ and $K^+$ through the channel, leading to plasma membrane depolarization facilitating, in many cell types, calcium uptake via other calcium-permeable channels[5,6]. TRPM4 activity regulates physiological processes such as cardiac conduction, smooth muscle contraction, insulin secretion, and immune responses[7–9]. Genetic defects in *TRPM4*

are found in patients with inherited cardiac conduction disorders. Furthermore, alterations of TRPM4 function are linked to diabetes, hypertension, and cancer[10–16]. Thus far, several small molecules identified and developed as potent and selective TRPM4 inhibitors are used as research tools to investigate the potential of TRPM4 as a therapeutic target and are promising candidates for drug development[17]. These molecules include 9-phenanthrol, flufenamic acid and the anthranilic acid derivatives CBA (4-chloro-2-[2-(2-chloro-phenoxy)-acetylamino]-benzoic acid) and NBA (4-chloro-2-(2-(naphthalene-1-yloxy) acetamido) benzoic acid[18–21]. NBA and CBA are thus far the most selective and potent TRPM4 inhibitors reported; however, it remains unknown how these drugs bind and inhibit TRPM4 activity[19]. To facilitate drug development targeting TRPM4, detailed structural information revealing the mode of binding these molecules to TRPM4 is essential.

Cryo-electron microscopy (cryo-EM) enabled the structure determination of several TRP channels. It revealed the druggable sites in these channels by structure determination in the presence of antagonists and drug molecules[3,17]. These structures have revealed different drug-binding pockets around the TRP channels'

[1]Laboratory of Biological Electron Microscopy, IPHYS, SB, EPFL, and Dept. Fundamental Microbiology, Faculty of Biology and Medicine, UNIL, Cubotron, Rt. de la Sorge, Lausanne, Switzerland. [2]Institute of Biochemistry and Molecular Medicine and Swiss National Centre of Competence in Research TransCure, University of Bern, Bern, Switzerland. [3]International Cancer Center, Shenzhen University Medical School, Shenzhen University, Shenzhen, Guangdong, China. ✉e-mail: henning.stahlberg@epfl.ch; hugues.abriel@unibe.ch

transmembrane domains (TMD). For example, the synthetic molecule icilin binds into a hydrophobic pocket of the Voltage Sensor Like Domain (VSLD) in TRPM8 and the binding of the small molecule inhibitor NDNA (N'-(3,4-dimethoxybenzylidene)-2-(naphthalen-1-yl) acetohydrazide) into the cavity between the S1-S4 domain and pore domain of TRPM5[22,23]. These two examples revealed different drug-binding sites within the same subfamily of TRP channels. Interestingly, lipid molecules can share binding sites with drug molecules and inhibit drug binding and activity, as demonstrated in the binding of 2-ABP (2-aminoethyl diphenylborinate) into the VBP (Vallanoid Binding Pocket) of TRPV2, which could be inhibited by binding an endogenous cholesterol molecule[24].

Structures of detergent-isolated TRPM4 studied by cryo-EM have revealed the binding sites of the cholesterol homolog CHS (Cholesteryl Hemisuccinate) used to purify TRPM4, suggesting potential endogenous cholesterol binding sites[25–28]. However, to date, no structures of TRPM4 have been reported to be bound to specific inhibitors. The addition of CHS required for stabilizing TRPM4 during detergent isolation could potentially occlude drug-binding sites, posing a challenge in understanding TRPM4 inhibition. In this work, we isolate TRPM4 in its native lipid environment using SMA (Styrene Maleic Anhydride) nanodiscs and determined its high-resolution structure in the presence and absence of the small molecules NBA and a derivative IBA (4-chloro-2-[2-(3-iodophenoxy)-acetylamino]-benzoic acid) which has a

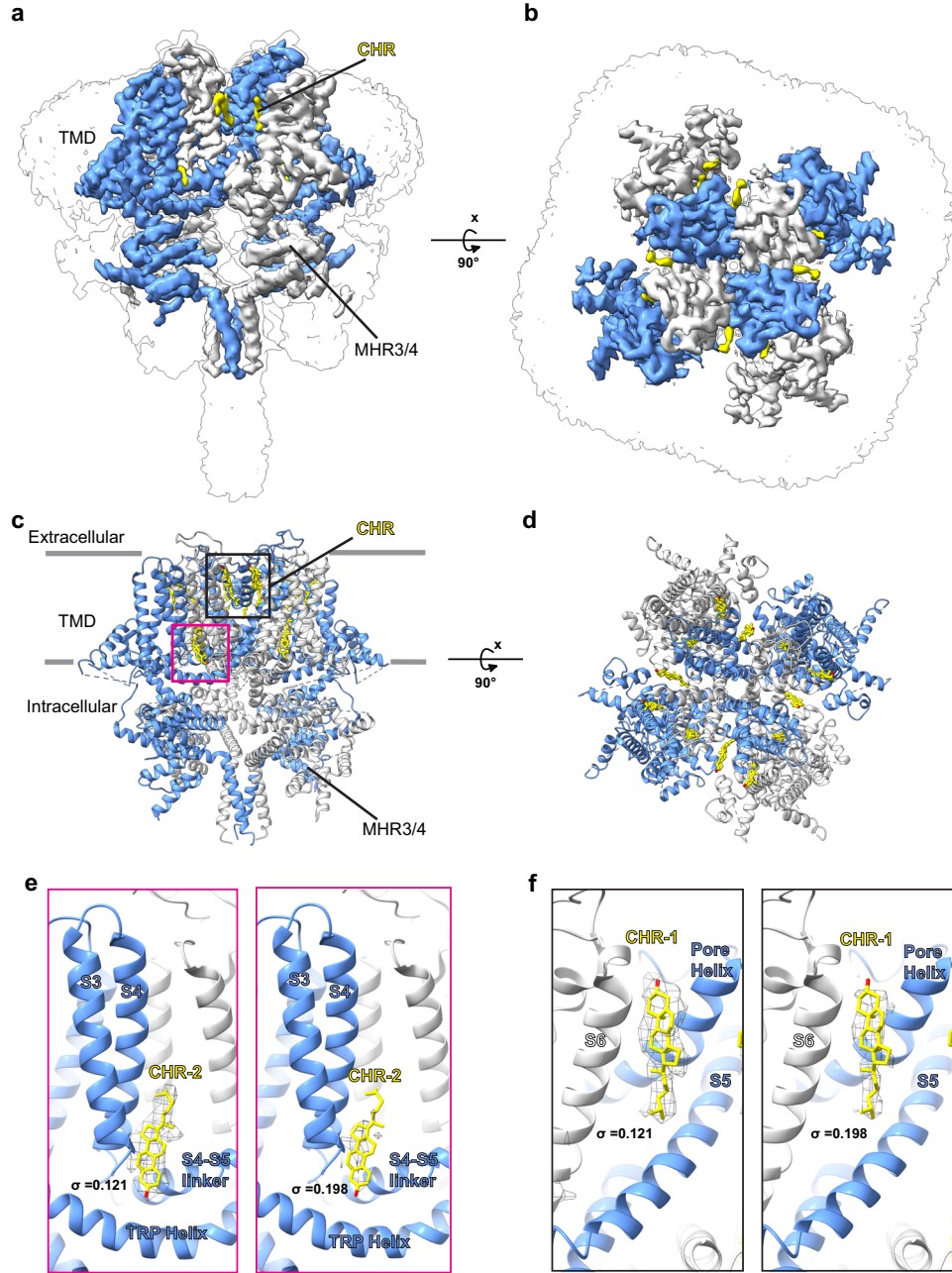

**Fig. 1 | Overall structure of HsTRPM4apo in native lipid nanodiscs. a, b** Two views of the cryo-EM densities of HsTRPM4apo in surface representation. Alternating protomers of TRPM4 are shown in blue and white. The TRPM homology regions (MHR3/4) are indicated. **c, d** Cartoon representation of the HsTRPM4apo corresponding to the cryo-EM densities in (**a, b**). The positions of bound cholesterol molecules (CHR) are indicated. The black box indicates the positions of CHR-1 and CHR-3, as shown in yellow. In contrast, the pink box indicates the position of CHR-2. **e** Cryo-EM density in mesh representation for CHR-2 (**f**) Cryo-EM density in mesh representation for CHR-1. In e and f, the density threshold level is indicated by σ.

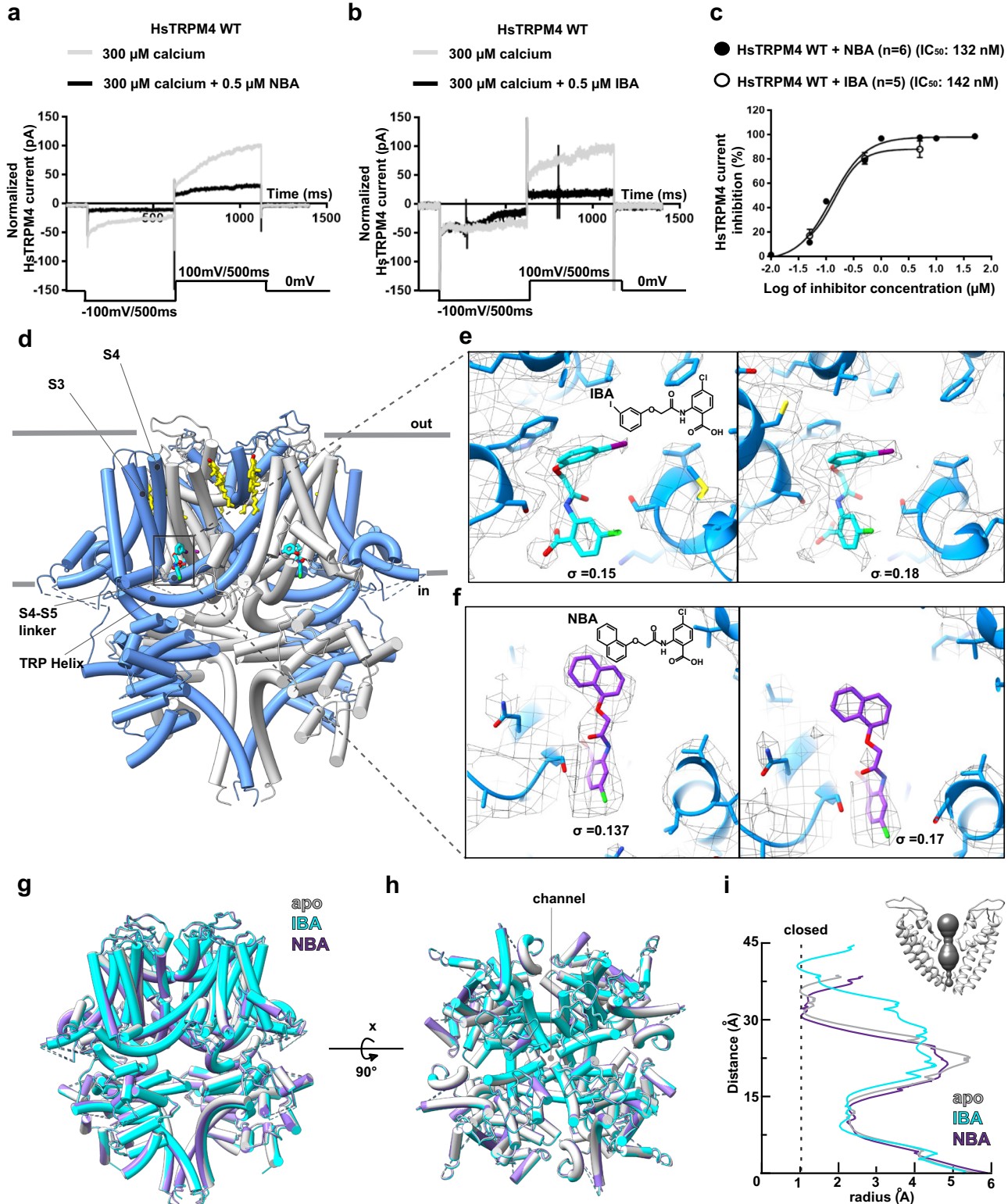

**Fig. 2 | Structures of HsTRPM4_NBA and HsTRPM4_IBA. a** Representative sodium current of the calcium-activated HsTRPM4 wildtype (WT) channel in the presence of a solution with 300 μM of free calcium (grey line) and in the presence of a solution with 300 μM of free calcium with 0.5 μM of NBA (black line). (**b**) in the presence of a solution with 300 μM of free calcium (grey line) and in the presence of a solution with 300 μM of free calcium with 0.5 μM of IBA (black line). **c** NBA and IBA dose-response curves of wildtype (WT). (n): number of cells. Data are presented as mean values +/− SEM. **d** The structure of inhibitor-bound HsTRPM4 with box and dashed lines indicating the inhibitor binding site. **e, f** Zoom-in of IBA and NBA with the cryo-EM density in mesh representation at two different density threshold levels as indicated by σ. **g, h** Superimposition of HsTRPM4_apo in light grey, HsTRPM4_NBA in purple and HsTRPM4_IBA in cyan. Two views of the structures are shown. **i** Pore radius for HsTRPM4_apo, HsTRPM4_NBA and HsTRPM4_IBA, all calculated using MOLE. In the top right corner are the pore-forming helices in HsTRPM4_apo. The pore profile is shown as a space-filling model (grey).

similar potency to NBA (Fig. 1; Fig. 2a–c and Supplementary Figs. 1–8; see method section for IBA synthesis)[29]. These structures reveal the mode of binding and inhibition of two potent and specific TRPM4 inhibitors, paving the way for future drug development and facilitating further investigations of TRPM4 as a therapeutic target for treating related maladies.

## Results

### Structures of HsTRPM4 in its native lipid environment

We determined cryo-EM structures of full-length human TRPM4 in endogenous lipid nanodiscs in apo (HsTRPM4apo), IBA (HsTRPM4IBA), and NBA (HsTRPM4NBA) bound states at overall resolutions of 3.7, 3.6, and 4.5 Å respectively (Figs. 1, 2; Supplementary Figs. 4–8; Table 1). All structures obtained were identical to previously determined cryo-EM structures of TRPM4 in detergent micelles and reconstituted nanodiscs[25–28]. However, we found that in native nanodiscs, densities for the cytosolic regions of TRPM4 are poorly resolved compared to previously determined structures, resulting in lower-resolution 3D reconstructions in these regions, which could be a result of TRPM4 solubilization using SMA. This effect was more pronounced in the cytosolic N-terminal TRPM homology regions 1 and 2 (MHR1/2), with the density in these regions much less defined (Supplementary Fig. 1c; Supplementary Fig. 2a, b). Therefore, 3D reconstructions could not be obtained for these regions (Supplementary Figs. 2; 4–6). These results suggest that SMA solubilization results in a less stable TRPM4 structure, which could result from the differences in lipid composition or binding compared with structures of TRPM4 in detergent micelles and reconstituted nanodiscs. Indeed, examination of the TMD revealed that although the structural arrangement of the transmembrane helices was identical, the lipid arrangement was strikingly different from previously determined TRPM4 structures[25–28]. Notably, the published structure of HsTRPM4 in a reconstituted lipid nanodisc, as well as cryo-EM density maps of detergent-solubilized human and mouse Trpm4 from this study, revealed ordered densities for annular phospholipids and three CHS molecules, one of which (CHS-2) occludes the drug binding site (Supplementary Figs. 2 and 3; Table 2)[26].

On the other hand, in our structures, ordered densities for annular lipids are markedly reduced, with primarily three densities for cholesterol molecules bound to TRPM4 observed (Fig. 1a–d; Supplementary Fig. 2a, b). Of the three, two cholesterol molecules (CHR-1 and CHR-2) bind in similar positions to CHS molecules in a previous structure in a reconstituted nanodisc (Fig. 1e, f; Supplementary Fig. 2a, b). One of these molecules (CHR-1) interacts with the S6 helix close to the channel pore, and another (CHR−2) in a pocket formed between S3 and S4 helices and the S4-S5 linker equivalent to the VBP of TRPV2; however, the density for this cholesterol molecule is weaker compared to CHR-1, suggesting lower occupancy (Fig. 1e, f). The third cholesterol molecule (CHR-3) in our structures binds between the S5 and S2 helix from an adjacent protomer differing from the previously reported position of CHS at the pre-S1 elbow (Supplementary Fig. 2a–c). The presence of less density for annular lipids in the native nanodiscs was an essential factor that enabled the assignment of density for the drugs in the HsTRPM4IBA and HsTRPM4NBA structures (Fig. 2d–i; Supplementary Fig. 8).

### The binding site of the small molecule inhibitors NBA and IBA in HsTRPM4

Analysis of the HsTRPM4IBA and HsTRPM4NBA structures revealed discrete densities for the IBA and NBA molecules located at the periphery of the inner membrane leaflet in the pocket formed between S3 and S4 helices, S4-S5 linker, and TRP helix (Fig. 2d–f). The presence of the ligands in this pocket suggests that these ligands can displace the endogenous cholesterol molecule (CHR-2) present in the HsTRPM4apo structure, which is possibly not tightly bound as judged by the weaker

**Table 1 | Cryo-EM data collection, refinement, and validation statistics for SMA solubilized HsTRPM4 samples**

| Data collection and processing | HsTRPM4apo (EMD-19057) (PDB 8RCR) | HsTRPM4IBA (EMD-19061) (8RCU) | HsTRPM4NBA (EMD-19069) (PDB 8RD9) (EMD-19060, local) |
|---|---|---|---|
| Nominal Magnification | 165 kx | 165 kx | 165 kx |
| Voltage (kV) | 300 | 300 | 300 |
| Recorded Micrographs | 6 429 | 7 500 | 11 851 |
| Electron exposure (e–/Å²) | 50 | 50 | 50 |
| Defocus range (μm) | 0.8–2.5 | 0.8–2.5 | 0.8–2.5 |
| Pixel size (Å) | 0.726 | 0.726 | 0.726 |
| Symmetry imposed | C4 | C1/C4 | C4 |
| Initial particle images (no.) | 1 719 323 | 1 149 090 | 1 805 182 |
| Final particle images (no.) | 16 308 | 20 345 | 15 954 |
| Map resolution (Å) | 3.67 | 3.59 | 4.50 |
| FSC threshold | 0.143 | 0.143 | 0.143 |
| Map resolution range (Å) | 30–3.3 | 30-3.2 | 30–3.6 |
| Map sharpening B factor (Å²) | −69.7 | −52.2 | −108.1 |
| **Refinement** | | | |
| Initial model used (PDB code) | 6BQV | 6BQV | 6BQV |
| Model composition | | | |
| Non-hydrogen atoms | 8 | 9 | 8 |
| Protein residues | 2420 | 2420 | 2422 |
| Ligands | CHR: 12 | IBA: 4 | NBA: 4 |
| | | CHR: 8 | CHR: 8 |
| *B* factors (Å²) | | | |
| Protein | 41.23/ 251.46/135.21 | 0.00/ 93.44/38.86 | 142.10/ 427.50/239.20 |
| Ligand | 63.19/ 103.12/79.00 | 15.44/ 136.15/47.75 | 20.00/ 33.04/22.07 |
| R.m.s. deviations | | | |
| Bond lengths (Å) | 0.012 (146) | 0.003 (0) | 0.004 (0) |
| Bond angles (°) | 0.801 (7) | 0.721 (0) | 0.925 (10) |
| Validation | | | |
| MolProbity score | 1.93 | 2.04 | 2.23 |
| Clashscore | 14.67 | 16.76 | 24.33 |
| Poor rotamers (%) | 0.00 | 0.19 | 0.05 |
| Ramachandran plot | | | |
| Favored (%) | 96.16 | 95.49 | 94.82 |
| Allowed (%) | 3.71 | 4.34 | 4.76 |
| Disallowed (%) | 0.13 | 0.17 | 0.42 |

density map for this molecule compared to other bound cholesterol molecules (Fig. 1e, f). Ligand binding induces only very subtle conformational changes in the structure of HsTRPM4, with the overall structures remaining broadly similar (Fig. 2g, h). The HsTRPM4 ion conduction pore is also in the closed state in all three structures; however, subtle changes in pore diameter are observed in the IBA-

**Table 2 | Cryo-EM data collection for DDM/CHS solubilized HsTRPM4 and MmTRPM4 samples**

| Data collection and processing | HsTRPM4 + Ca$^{2+}$ | HsTRPM4+NBA+Ca$^{2+}$ | MmTRPM4 + Ca$^{2+}$ |
|---|---|---|---|
| Nominal Magnification | 96kx | 96kx | 165kx |
| Voltage (kV) | 300 | 300 | 300 |
| Recorded Micrographs | 11 877 | 10 219 | 8 313 |
| Electron exposure (e–/Å2) | 40 | 40 | 60 |
| Defocus range (μm) | 0.8-2.5 | 0.8-2.5 | 0.8-2.5 |
| Pixel size (Å) | 0.83 | 0.83 | 0.726 |
| Symmetry imposed | C4 | C4 | C4 |
| Initial particle images (no.) | 4 430 540 | 2 774 028 | 4 340 237 |
| Final particle images (no.) | 128 316 | 40 061 | 100 227 |
| Map resolution (Å) | 3.04 | 2.98 | 2.87 |
| FSC threshold | 0.143 | 0.143 | 0.143 |
| Map resolution range (Å) | 30-2.6 | 30-2.5 | 30-2.5 |

bound structure, although it remained closed (Fig. 2i). In this study, exogenous calcium was not added to the SMA-extracted TRPM4 samples used for cryo-EM because adding divalent ions destabilizes SMA nanodiscs[29]. However, our cryo-EM density maps of TRPM4 in native nanodiscs revealed a strikingly apparent density in the calcium-binding site as observed in structures of detergent-extracted human and mouse TRPM4 supplemented with Ca$^{2+}$. Indeed, upon fitting a previously determined structure of calcium-bound TRPM4 into our cryo-EM map of SMA-extracted TRPM4, we observe that the calcium ion together with its interacting residues Glu828, Gln831, Asp 868, and Arg905 fit very well into our cryoEM density of SMA extracted TRPM4 similar to cryoEM density maps of calcium supplemented detergent isolated TRPM4 which were reproduced in this study (Supplementary Fig. 9). Indeed, this suggests the binding of endogenous Ca$^{2+}$ ions in the SMA extracted TRPM4. Still, we cannot rule out the possibility of other endogenously derived ions occupying the binding site. Nonetheless, the occupation of the calcium-binding site in our structures did not result in an open state of the channel like previously described in calcium-bound TRPM4 structures.

Both IBA and NBA share a common anthranilic acid moiety but differ in that NBA possesses an additional ring of its naphthalene substituent, making it bulkier and more hydrophobic than the smaller 3-iodophenyl ring of IBA (Fig. 2e, f, Supplementary Fig. 8). The additional ring of the naphthalene substituent of NBA was an important fiducial marker for the overall placement of the ligands in the density maps. In both HsTRPM4$_{IBA}$ and HsTRPM4$_{NBA}$ structures, the hydrophilic anthranilic acid moiety containing acidic, amide, and chloride groups faces towards the cytosol interacting with charged amino acid side chains from TRPM4 (Fig. 3a, b). Notably, the anthranilic acid moiety interacts with His908 from the S4 helix, Tyr1057, Gln1061, and Arg1064 from the TRP helix, as well as with Ser924 from the S4-S5 linker, and Ser863 from the S3 helix in both structures (Fig. 3a, b). These interactions between the anthranilic acid moiety and residues of TRPM4 reveal the chemical basis behind the specific binding of the anthranilic acid derivatives to TRPM4.

NBA and IBA have a similar potency for TRPM4 inhibition, possibly by making additional unknown interactions in the binding pocket (Fig. 2a–c). Our structures show that the naphthalene substituent and the 3-iodophenyl ring of NBA and IBA bind into a hydrophobic pocket between the S3 and S4 helices of TRPM4 (Fig. 3c, d; Supplementary Fig. 7d, e), making hydrophobic interactions with the residues that decorate this pocket, including Val901, Val904 and Leu907 (Fig. 3c, d). Therefore, these structures reveal a rationale for the potency and specificity of NBA and IBA and pave the way for designing chemical modifications of these molecules to enable increased targeting of the binding pocket to develop more potent and specific inhibitors.

## Validation of the NBA and IBA binding site in HsTRPM4

To disrupt the drug-protein interactions in the hydrophilic region of the drug binding site, we introduced the HsTRPM4 mutations Ser863Ala (S863A), His908Ala (H908A), Ser924Ala (S924A), Lys925Ala (K925A), Tyr1057Ala (Y1057A), Gln1061Ala (Q1061A), Arg1064Ala (R1064A), Arg1064Gly (R1064G), Arg1064Ser (R1064S), and the double variant Ser863Ala/Ser924Ala (S863A/S924A). Additional Val901Trp (V901W) and Val904Trp (V904W) are space-filling mutations that disrupt the hydrophobic pocket. These mutations were introduced to reduce the drug's inhibitory effect compared to wildtype HsTRPM4 (Fig. 3a–d). First, the expression of the different variants was investigated using the western blot approach (Supplementary Fig. 10). Only Ser863Ala and Gln1061Ala variants showed a significantly reduced expression compared to the wildtype channel (Supplementary Fig. 10b). In parallel, functional experiments revealed that after activation of the HsTRPM4 function by adding a solution with a concentration of 300 μM of free calcium, almost all variants displayed a significant reduction in the sodium current of the calcium-activated HsTRPM4 channel except Ser863Ala, Ser924Ala, Arg1064Ala, and Arg1064Gly quantify at a membrane voltage of +100 mV (Fig. 3e, f, Supplementary Fig. 12). Due to the reduction of activation in these mutants and the proximity of these residues in the NBA binding site to the calcium-binding site (Fig. 4a), we wondered if this decrease in function observed may be due to an alteration of the calcium sensitivity of those variants. The calcium sensitivity curves performed on some loss-of-function HsTRPM4 variants (Val904Trp, Tyr1057Ala, Gln1061Ala, Arg1064Ser, and the double variant Ser863Ala/ Ser924Ala) show that the different EC$_{50s}$ for calcium are not higher compared to wildtype HsTRPM4 suggesting that this decrease of function is not due to a reduction of calcium sensitivity (Fig. 4b; Supplementary Fig. 13a). On the contrary, the different EC$_{50s}$ of the variants are smaller than the wildtype EC$_{50}$, suggesting an increase in calcium sensitivity (Fig. 4b; Supplementary Fig. 13a). Following those experiments and knowing that the inhibitory effect mediated by NBA will be investigated on these variants in the presence of calcium (to activate them), control experiments were performed using HsTRPM4 wildtype channels to explore the potential influence of NBA on calcium sensitivity. As for the variants, the results suggest that NBA does not decrease the calcium sensitivity of the wildtype channels (Fig. 4b; Supplementary Fig. 13b). The absence of a decrease of the calcium sensitivity in the presence of NBA and amino acid modifications led us to perform the final experiment investigating the consequences of modifying those amino acids on the inhibitory effect mediated by NBA. As a proof of concept, the mutations Ser863Ala, Val901Trp, His908Ala, Ser924Ala, Arg1064Ala, Arg1064Gly, and the double variant Ser863Ala/ Ser924Ala were investigated. Compared to wildtype HsTRPM4, NBA dose-effect curves on single variants of the HsTRPM4 channel show a drastic increase (up

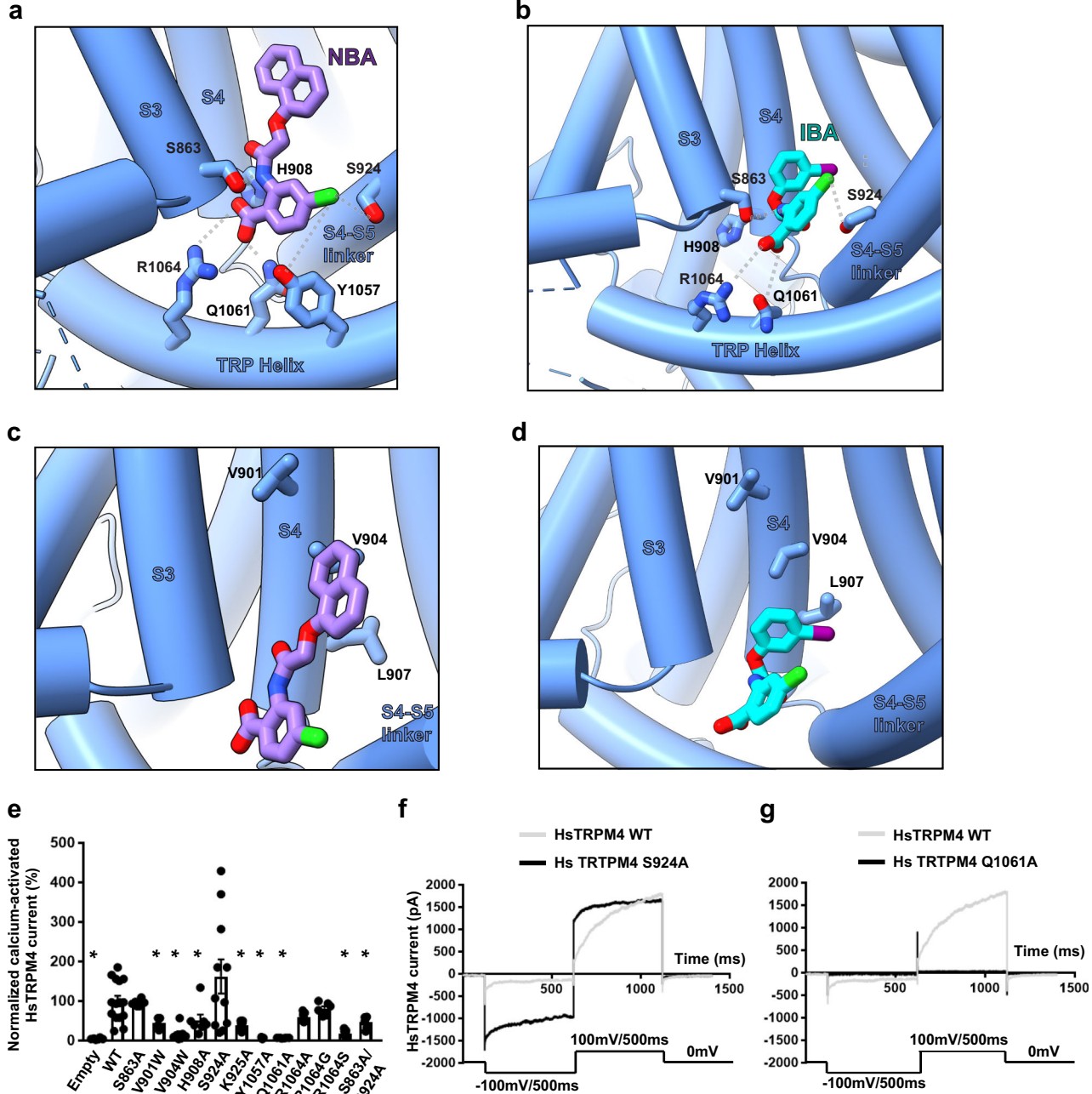

**Fig. 3 | Amino acid interactions in the drug binding pocket. a** Amino acid residues of HsTRPM4 involved in interactions with the anthranilic acid moiety of NBA (in purple) and (**b**) IBA (cyan) are shown. **c** Amino acid residues of HsTRPM4 involved in hydrophobic interactions with NBA (in purple) and (**d**) IBA (in cyan) are shown. **e** Graphical plot of normalized recorded sodium current of the calcium-activated HsTRPM4 channel for the wildtype (WT) and HsTRPM4 variants. Data are presented as mean values +/− SEM. (*) represents the *p*-value < 0.05 compared to the wildtype condition. The number of cells per condition recorded is: Empty (*n* = 6), HsTRPM4 WT (*n* = 15), HsTRPM4 S863A (*n* = 6), HsTRPM4 V901W (*n* = 6), HsTRPM4 V904W (*n* = 7), HsTRPM4 H908A (n = 7), HsTRPM4 S924A (*n* = 11), HsTRPM4 K925A (*n* = 7), HsTRPM4 Y1057A (*n* = 7), HsTRPM4 Q1061A (*n* = 6), HsTRPM4 R1064A (*n* = 7), HsTRPM4 R1064G (*n* = 6), HsTRPM4 R1064S (*n* = 7), and HsTRPM4 S863A/S924A (*n* = 6). The Mann-Whitney test, which is two-tailed, has been used to compare Hs TRPM4 to another condition, and multiple comparisons

have been performed. The *p*-values are: HsTRPM4 WT versus Empty (*p* < 0.0001), HsTRPM4 WT versus HsTRPM4 S863A (*p* = 0.9852), HsTRPM4 WT versus HsTRPM4 V901W (*p* = 0.0057), HsTRPM4 WT versus HsTRPM4 V904W ((*p* < 0.0001), HsTRPM4 WT versus HsTRPM4 H908A (*p* = 0.02), HsTRPM4 WT versus HsTRPM4 S924A (*p* = 0.4495), HsTRPM4 WT versus HsTRPM4 K925A (*p* = 0.0022), HsTRPM4 WT versus HsTRPM4 Y1057A (*p* < 0.0001), HsTRPM4 WT versus HsTRPM4 Q1061A (*p* < 0.0001), HsTRPM4 WT versus HsTRPM4 R1064A (*p* = 0.0528), HsTRPM4 WT versus HsTRPM4 R1064G (*p* = 0.6085), HsTRPM4 WT versus HsTRPM4 R1064S (*p* < 0.0001), and HsTRPM4 WT versus HsTRPM4 S863A/S924A (*p* = 0.0133). **f, g** Representative traces of wildtype (WT) and variants sodium currents of the calcium-activated HsTRPM4 channels: S924A and Q1061A. Sodium currents of the calcium-activated HsTRPM4 variant channels are generally smaller than WT currents, as depicted in (**e**), due to the amino acid alteration.

to one log of difference) of the IC$_{50s}$ for the NBA, suggesting a decreased efficiency of inhibition by NBA (Fig. 5a). Compared to the wildtype construct, but also to single amino acid variant Ser863Ala and Ser924Ala, a more pronounced decrease in the efficiency of NBA of up

to two logs of difference is observed in the presence of the double mutation Ser863Ala/ Ser924Ala (Fig. 5a, c, d; Supplementary Fig. 14). In addition, a control experiment was performed using a mutant of HsTRPM4 linked to cardiac dysfunction, which has already been

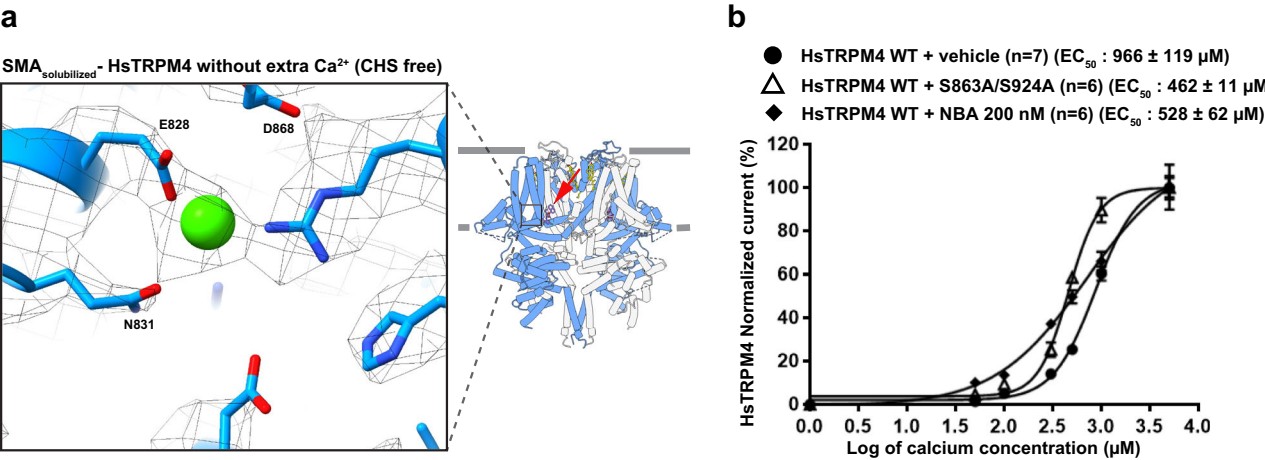

**a**

SMA$_{solubilized}$ - HsTRPM4 without extra Ca$^{2+}$ (CHS free)

E828  D868  N831

**b**

● HsTRPM4 WT + vehicle (n=7) (EC$_{50}$ : 966 ± 119 µM)
△ HsTRPM4 WT + S863A/S924A (n=6) (EC$_{50}$ : 462 ± 11 µM)
◆ HsTRPM4 WT + NBA 200 nM (n=6) (EC$_{50}$ : 528 ± 62 µM)

**Fig. 4 | Calcium sensitivity of TRPM4 variants. a** Zoom-in of the calcium-binding site with the cryo-EM density in mesh representation. Green circle indicates the putative binding location of Ca$^{2+}$ ion. The inhibitor binding site is also indicated with a red arrow. **b** Normalized calcium sensitivity curves for wildtype (WT) HsTRPM4 and loss-of-function HsTRPM4 variant S863A/S924A, and the effect of a solution containing the compounds NBA (200 nM) on this calcium sensitivity. Data are presented as mean values +/− SEM. (n): number of cells.

reported in the literature: the mutant HsTRPM4 T677I[30]. This variant was selected based on its presence outside the putative NBA binding pocket and because this point mutation did not alter the biophysical properties of channels[30]. This mutation did not alter the IC$_{50}$ of the NBA, as observed with the NBA binding pocket variants (Fig. 5e, f). Overall, these data strongly support the notion that those amino acids of HsTRPM4 are part of the binding pocket for NBA or IBA, as determined by our cryo-EM structures.

## Discussion

We have utilized cryo-EM to determine the high-resolution structures of HsTRPM4 bound to specific inhibitors. Importantly, we find that using SMA to isolate HsTRPM4 in its endogenous lipid nanodisc was necessary to assign densities for the inhibitor molecules, which electrophysiology experiments could further validate. This finding supports the relevance of previous attempts to isolate TRPM4 in its native lipid environment but, unfortunately, yielded lower-resolution structures[31]. High-resolution structures of TRPM4 have previously been determined in detergent micelles and in reconstituted lipid nanodiscs, both of which require detergent isolation of the protein from the cell and, by so doing, introduce artifacts to the lipid environment of the protein. An essential factor is that the added CHS, to maintain protein stability in detergent extraction, binds in specific sites in the protein, which may or may not represent actual cholesterol-binding sites. We observed apparent differences in the binding arrangement of annular lipids to TRPM4 from the structural data when isolated with SMA. Most striking is that a CHS molecule completely occupies the drug binding site, suggesting that obtaining a structure or performing biochemical experiments with detergent-CHS isolated protein and these inhibitors does not represent the endogenous scenario. Indeed, identifying the inhibitor binding site was ambiguous in our previous attempts to determine the structure of detergent-isolated TRPM4 in a complex with NBA or IBA. This finding also shows the importance of the composition and binding arrangement of the lipid annulus to the function and inhibition of the target membrane protein. This study shows that using SMA would be advantageous for the structure determination of drugs with membrane proteins, particularly in cases where the drug binds in the TMD exposed to lipid molecules.

During the revision of this work, another paper was published, which revealed the importance of temperature in modulating the structure of detergent-extracted TRPM4. More precisely, at the warm temperature of 37 °C, TRPM4 was shown to undergo structural rearrangements, particularly in MHR1/2 domains of the ICD; interestingly, this region is also flexible and poorly resolved in our structures, which could suggest an interplay of the lipid environment with the temperature-induced structural movement[32]. Furthermore, the binding site of the allosteric modulator DVT was also shown to change in a temperature-dependent manner, leading to channel opening via the relative movement of the TRP helix to S1-S4 helices. This movement of the TRP helix is required for the activation of the TRPM2 channel and is also required for desensitization of the TRPM8 channel[32–34]. These findings reveal the important role of TRP helix movement in regulating the channel pore of the TRPM subfamily. Based on our results and these previous studies, we propose that the binding of NBA or IBA to TRPM4 may restrict the movement of the TRP helix, maintaining the TRPM4 channel pore in a closed state even in the presence of activation stimuli (Fig. 6). The role of the bound cholesterol molecule in the same pocket remains unclear. This cholesterol could also regulate channel function in response to changes in the membrane environment.

A major limitation in our study is that we could not determine structures of SMA-extracted TRPM4 in the presence of exogenous Ca$^{2+}$ ions, as the addition of calcium destabilizes SMA nanodiscs. Although we did observe density in the calcium-binding site, we cannot rule out an effect or interrelation between the binding of Ca$^{2+}$ ions and NBA or IBA. Moreover, we did not observe decreases in calcium sensitivity in the presence of NBA, but the implications of which remain unclear (Fig. 4). However, other published structures of TRPM4 in the presence and absence of Ca$^{2+}$ ions reveal that the NBA or IBA binding site and residues involved in the interaction between TRPM4 and NBA or IBA remain accessible for binding of a CHS molecule in both states suggesting this pocket is accessible for binding of the inhibitors in both states[26].

Our study has identified a binding pocket for inhibiting TRPM4 by the anthranilic acid derivatives (Fig. 2d–f). These drugs bind in a pocket between S3 and S4 helices, S4-S5 linker, and TRP helix. However, detailed insights into the mechanism of inhibition will require structure determination of TRPM4 in its open state, which has been elusive. Adding Ca$^{2+}$ ions is insufficient to obtain the open state with purified TRPM4 in lipid nanodiscs. Still, the presence of a membrane potential may be necessary. In the future, we will seek to address this by studying TRPM4 in lipid vesicles. Nonetheless, identifying a binding pocket for TRPM4 inhibitors presents a significant milestone towards

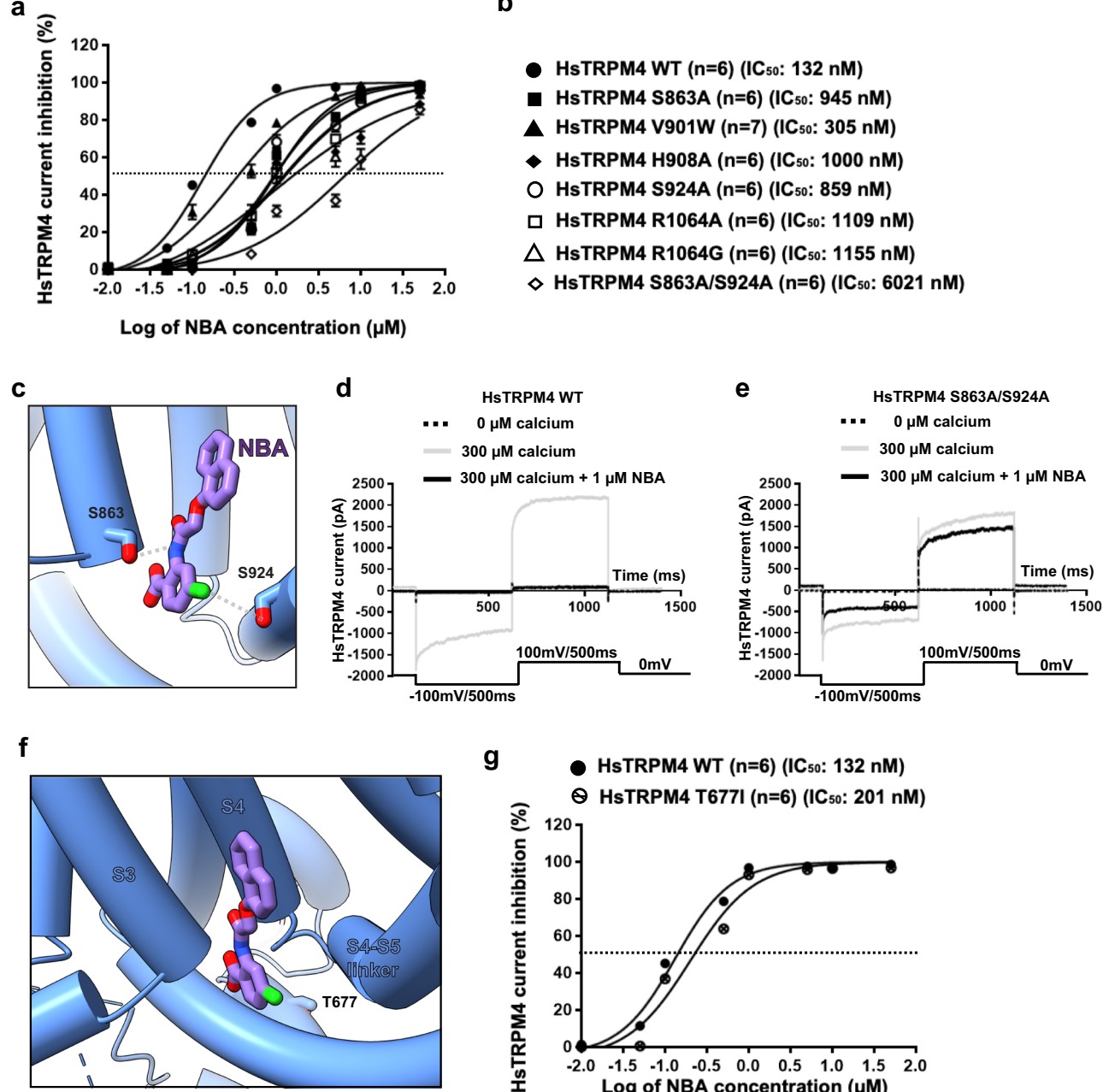

**Fig. 5 | Effect of HsTRPM4 variants on NBA inhibition. a, b** NBA dose-response curves of wildtype (WT) and variants of the predicted binding pocket of the HsTRPM4 channel, as a function of the decadic logarithm of the NBA concentration. Data are presented as mean values +/− SEM. (n): number of cells. **c** Locations of S863 and S924 in the drug binding pocket (**d, e**) Representative traces of wildtype (WT) and variant HsTRPM4 sodium currents: S863A/S924A in the absence of calcium (dotted black line), in the presence of a solution with a concentration of 300 μM of free calcium (grey line) and in the presence of a solution with a concentration of 300 μM of free calcium and 1 μM of NBA (black line). **f** Representation of the T677I variant located outside of the drug binding pocket of the TRPM4 channel. **g** NBA dose-response curves of wildtype (WT) and a T677I variant. (n): number of cells.

further developing potent and even more specific drugs targeting the TRPM4 ion channel for therapeutic intervention.

## Methods

### Synthesis of 4-Chloro-2-(2-(3-iodophenoxy)acetamido)benzoic acid (IBA)

**General Remarks.** Reagents and organic solvents were purchased from commercial suppliers and used without further purification. Deionized water produced in-house or commercially available Milli-Q® water was used depending on the application. Aqueous solutions of sodium hydroxide, hydrogen chloride, saturated ammonium chloride, and saturated sodium chloride (brine) were prepared with deionized water. Thin layer chromatography (TLC) was performed using Macherey-Nagel ALUGRAM® Xtra SIL G/UV254 plates coated with 0.20 mm silica gel 60 containing fluorescent indicator. High-pressure liquid chromatography (HPLC) was performed using a Thermo Fisher Scientific UltiMate 3000 RSLCnano System composed of a DIONEX UltiMate 3000 Pump, a DIONEX UltiMate 3000 Sampler, a DIONEX UltiMate 3000 Column Compartment, and a DIONEX UltiMate 3000 Diode Array Detector. HPLC measurements were conducted using Milli-Q® water (+ 0.1% TFA) and acetonitrile (+ 0.1% TFA) as eluents and an Acclaim™ 120 C18 column (Thermo Scientific™). Flash column

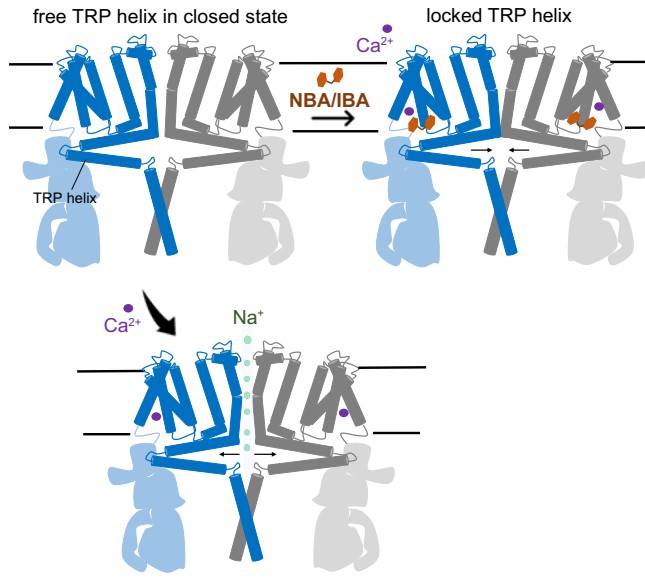

free TRP helix in closed state locked TRP helix

NBA/IBA

TRP helix

$Ca^{2+}$ $Na^{+}$

free TRP helix in open state

**Fig. 6 | Proposed inhibition mechanism.** The binding of anthranilic acid drugs such as NBA, CBA, and IBA in the TRPPM4 drug binding site leads to channel blocking, possibly by restricting the movement of the TRP helix, thereby preventing channel opening upon calcium activation.

chromatography (LC) was performed using the Teledyne Isco Combi*Flash*©*Rf*+ system. Teledyne Isco Redi*Sep*©*Rf* dry load cartridges were used to prepare dry loads. If not stated otherwise, dry loads were prepared on silica gel. Teledyne Isco Silica Redi*Sep*©*Rf* prepacked silica flash columns of two sizes (24 g and 80 g) were used. Nuclear magnetic resonance spectroscopy (NMR) was performed at the Department of Chemistry, Biochemistry, and Pharmaceutical Sciences, Universität Bern (Furrer Group) using a Bruker AVANCE III HD 300 GA spectrometer with a magnetic field of 7.05 Tesla and operating frequencies of 300.13 MHz for $^{1}$H measurements and 75.48 MHz for $^{13}$C measurements. High-resolution mass spectrometry (HRMS) was performed by the mass spectrometry service (Schürch group) at the Department of Chemistry, Biochemistry and Pharmaceutical Sciences, Universität Bern. The measurements were performed using electrospray ionization (ESI) and a ThermoScientific LTQ Orbitrap XL mass spectrometer with high mass resolution (m/Δm > 100'000) and accuracy (Δm <3ppm). The previously published synthetic procedure (Ozhathil et al., 2018) to generate similar anthranilic anilide compounds was slightly adapted.

**Methyl 4-chloro-2-(2-chloroacetamido) benzoate.** Methyl 2-amino-4-chlorobenzoate (1.7876 g, 9.6308 mmol) and potassium carbonate (2.6620 g, 19.2616 mmol, 2 eq.) were dissolved in tetrahydrofuran (150 mL) and stirred for 10 min at room temperature. Not all potassium carbonate dissolved completely. The mixture was then cooled in an ice bath, and chloroacetyl chloride (1.0919 g, 9.6676 mmol, 0.77 mL, 1 eq.) was added dropwise via a syringe. This mixture was stirred at 0 °C for 10 min and further stirred at room temperature for 16 h. The reaction mixture showed a pale pink color and was monitored by thin layer chromatography (eluent: cyclohexane/ethyl acetate, 4:1). After full conversion of the starting material, water was added to the reaction mixture and the product was extracted with ethyl acetate. The combined organic phases were then washed with brine, dried over magnesium sulfate, filtered through celite, and the volatiles were evaporated under reduced pressure. The crude pale-yellow and solid product was purified by flash column chromatography (eluent: cyclohexane/ethyl acetate, gradient from 0% to 20% ethyl acetate). The

product (white powder) was dried *in vacuo*. Yield Methyl 4-chloro-2-(2-chloroacetamido) benzoate quant., 2.5 g, 9.5387 mmol. $^{1}$H NMR (300 MHz, DMSO-$d_6$) δ 11.44 (s, 1H), 8.52 (d, $J$ = 2.2 Hz, 1H), 8.00 (d, $J$ = 8.6 Hz, 1H), 7.33 (dd, $J$ = 8.6, 2.2 Hz, 1H), 4.48 (s, 2H), 3.89 (s, 3H). $^{13}$C NMR (75 MHz, DMSO-$d_6$) δ 166.75, 165.67, 140.29, 138.75, 132.51, 123.77, 119.84, 115.57, 52.82, 43.35. HRMS (ESI) $m/z$ [M + H]$^{+}$ calculated for $C_{10}H_9Cl_2NO_3$ 262.0032, found 262.0037.

**Methyl 4-chloro-2-(2-(3-iodophenoxy)acetamido)benzoate.** Methyl 4-chloro-2-(2-chloroacetamido) benzoate (0.3270 g, 1.2475 mmol) was dissolved in dimethylformamide (6 mL), and potassium carbonate (0.3465 g, 2.5073 mmol, 2 eq.) was added. This mixture was stirred for 10 min at room temperature. Not all potassium carbonate dissolved completely. 3-Iodophenol (0.3019 g, 1.3723 mmol, 1.1 eq.) was then dissolved in dimethylformamide (6 mL) and added via syringe to the reaction mixture, which was stirred at 80 °C. The color of the reaction mixture changed to deep brown. The reaction was monitored by thin layer chromatography (eluent: cyclohexane/ethyl acetate, 4:1). Upon full conversion of the starting material after 60 min, water was added to the mixture, resulting in a brown and cloudy suspension. The product was extracted with ethyl acetate, and the combined organic phases were then washed with brine, dried over magnesium sulfate, filtered through celite, and the volatiles were evaporated under reduced pressure. The crude product was purified by flash column chromatography (eluent: cyclohexane/ethyl acetate, gradient from 0% to 20% ethyl acetate), and the purified product (white powder) was dried *in vacuo*. Yield Methyl 4-chloro-2-(2-(3-iodophenoxy)acetamido) benzoate 84%, 0.4653 g, 1.0442 mmol. $^{1}$H NMR (300 MHz, DMSO-$d_6$) δ 11.76 (s, 1H), 8.70 (d, $J$ = 2.1 Hz, 1H), 8.01 (d, $J$ = 8.6 Hz, 1H), 7.48 (t, $J$ = 1.9 Hz, 1H), 7.40 (ddd, $J$ = 6.4, 2.5, 1.6 Hz, 1H), 7.30 (dd, $J$ = 8.6, 2.2 Hz, 1H), 7.15 – 7.09 (m, 2H), 4.79 (s, 2H), 3.91 (s, 3H). $^{13}$C NMR (75 MHz, DMSO-$d_6$) δ 167.24, 166.87, 157.61, 140.57, 138.94, 132.56, 131.50, 130.66, 123.67, 123.34, 119.33, 114.75, 114.64, 95.04, 67.35, 52.76. HRMS (ESI) $m/z$ [M + H]$^{+}$ calculated for $C_{16}H_{13}ClINO_4$ 445.9651, found 445.9647.

**4-Chloro-2-(2-(3-iodophenoxy)acetamido)benzoic acid (IBA).** Methyl 4-chloro-2-(2-(3-iodophenoxy)acetamido)benzoate (0.4531 g, 1.0168 mmol) was dissolved in methanol (100 mL) and potassium hydroxide (0.1744 g, 3.1075 mmol, 3 eq.) was dissolved in Milli-Q® water (20 mL). The aqueous potassium hydroxide solution was added to the reaction mixture, which was stirred at 65 °C, and progress was monitored by thin layer chromatography (eluent: dichloromethane/methanol, 9:1). Upon full conversion of the starting material after 30 min, the reaction mixture was cooled down to room temperate. Aqueous hydrochloric acid solution (1 M, ca. 100 mL) was then added, resulting in immediate precipitation of the product, which was filtered off using a glass filter frit (Por. 4), and the solid white product was dried *in vacuo*. Yield 4-Chloro-2-(2-(3-iodophenoxy)acetamido)benzoic acid 79%, 0.3466 g, 0.8030 mmol. Purity (HPLC): 98.52% (at 254 nm). $^{1}$H NMR (300 MHz, DMSO-$d_6$) δ 14.08 (s, 1H), 12.25 (s, 1H), 8.77 (d, $J$ = 2.1 Hz, 1H), 8.03 (d, $J$ = 8.6 Hz, 1H), 7.46 (m, 1H), 7.38 (dt, $J$ = 7.0, 1.7 Hz, 1H), 7.27 (dd, $J$ = 8.6, 2.2 Hz, 1H), 7.19 – 7.08 (m, 2H), 4.77 (s, 2H). $^{13}$C NMR (75 MHz, DMSO-$d_6$) δ 168.78, 167.24, 157.58, 141.12, 138.69, 133.05, 131.48, 130.55, 123.74, 123.06, 118.78, 114.94, 114.29, 95.06, 67.21. HRMS (ESI) $m/z$ [M-H]$^{-}$ calculated for $C_{15}H_{11}ClINO_4$ 429.9349, found 429.9348.

### Description of drugs

Stock solutions of 4-chloro-2-(2-(naphthalene-1-yloxy)acetamido)benzoic acid (NBA) and the iodo-modified congener; 4-chloro-2-(2-(3-iodophenoxy)acetamino)benzoic acid (IBA) were dissolved in 100% DMSO to a final concentration of 10 mM. IBA was synthesized to increase the electron densities when using the cryo-EM approaches. Although, as proof of concept, only NBA has been used in the

functional experiments presented in this study, the inhibitory efficacity of IBA has been evaluated and is the same as of NBA. NBA and IBA solutions were freshly made before each experiment. The lipophilicity of these drugs (cLog *P*) has been calculated using the algorithms on Swiss ADME from the Swiss Institute of Bioinformatics (http://www.swissadme.ch/). The Log *P* corresponds to the compound's concentration ratio at equilibrium between organic (octanol) and aqueous phases. A negative Log *P* means the compound is hydrophilic, and a positive value for Log *P* denotes a more lipophilic compound. At pH = 7.4, the NBA and IBA compounds (both protonated (carboxylic acid) and deprotonated (carboxylate) forms) tend to be lipophilic. NBA is slightly more lipophilic than IBA (consensus cLog *P* protonated/deprotonated: NBA: 3.59/3.32; IBA: 3.25/3.05). The decadic $\log_{10}$ is used here.

### Plasmid constructs design and cloning
For this study, the wild-type codon-optimized *H. sapiens* (Hs) and *M. musculus* (Mm) TRPM4 genes coding full-length TRPM4 were synthesized, fused to consecutive C-terminal HA (hemagglutinin)−and FLAG-tags, and cloned into a pCDNA-3.1 plasmid for expression in HEK293 cells by the CMV (cytomegalovirus) promoter (GenScript Biotech). Mutant variants were generated by site-directed mutagenesis (GenScript Biotech).

### Protein expression and purification
The full-length *H. sapiens* TRPM4 and *M. musculus* TRPM4 were expressed and purified from HEK293F cells grown in suspension. For expression, HEK293F cells were transfected with 1 mg of a plasmid containing the TRPM4 gene per liter of cells using PEI (polyethylenimine). The cultures were grown at 37 °C and 5% $CO_2$ for 48 h. The cultures were harvested by centrifugation at 3000 x g for 30 min at 4 °C and washed in 1x PBS, followed by another round of centrifugation. The pellets were carefully resuspended in lysis buffer containing HEPES-NaOH, pH 7.5, 200 mM NaCl and supplemented with cOmplete™ EDTA-free Protease Inhibitor Cocktail (Roche). 4 tablets of protease inhibitor cocktail were added per 100 ml of buffer. Following resuspension, the cells were lysed by sonication for a total of 1 min in 10 s On and OFF cycles and an amplitude of 35%. After sonication, the membrane fraction was harvested by centrifugation using an Optima XPN-100 ultracentrifuge (Beckman Coulter) with the Ti45 rotor and spun at $70,560 \times g$ for 30 min at 4 °C. The resulting pellets were stored at −80 °C. For solubilization of the membrane fraction with SMALP200, 12 g of pellet was resuspended in 30 ml of solubilization buffer containing HEPES-NaOH, pH 7.5, 200 mM NaCl and 0.5% SMALP200 (CubeBiotech GmbH) supplemented with 2 tablets of cOmplete™ EDTA-free Protease Inhibitor Cocktail (Roche). The resuspended pellet was homogenized manually in a 40 ml Kimble glass homogenizer (Sigma). The tube containing the mixture was then placed in a bottle with a stir bar and left to incubate at 4 °C for 1 h 30 min. The homogenate was clarified by centrifugation for 30 min at $70,560 \times g$ at 4 °C in an Optima XPN Ultracentrifuge (Beckman Coulter) using a Ti-45 rotor. The supernatant, which contained soluble FLAG-tagged HsTRPM4 mixed with 1 ml of Anti-FLAG® M2 affinity gel (Millipore, Billerica, MA) pre-equilibrated with wash buffer containing 25 mM HEPES-NaOH, pH 7.5, 200 mM NaCl. The beads were washed with 100 mL of wash buffer containing 25 mM HEPES-NaOH, pH 7.5, 200 mM NaCl and eluted with 4 mL elution buffer containing 25 mM HEPES-NaOH, pH 7.5, 200 mM NaCl, 120 μg/ml of 3xFLAG peptide, followed by concentration on 100 K Amicon Ultra-15 concentrators (Millipore, Billerica, MA) to an absorbance at 280 nm of 1.0 to prepare cryo-EM grids.

For detergent solubilization, 12 g of pellet was resuspended in 30 ml of solubilization buffer containing HEPES-NaOH, pH 7.5, 200 mM NaCl, 1% n-Dodecyl-beta-Maltoside (DDM) and 0.1% Cholesteryl Hemissucinate (CHS)

(Avanti Pola lipids) supplemented with 2 tablets of cOmplete™ EDTA-free Protease Inhibitor Cocktail (Roche). The resuspended pellet was homogenized manually in a 40 ml Kimble glass homogenizer (Sigma). The tube containing the mixture was then placed in a bottle with a stir bar and left to incubate at 4 °C for 2 h. The homogenate was clarified by centrifugation for 30 min at $70,560 \times g$ at 4 °C in an Optima XPN Ultracentrifuge (Beckman Coulter) using a Ti-45 rotor. The supernatant, which contains soluble FLAG-tagged HsTRPM4 mixed with 1 ml of Anti-FLAG® M2 affinity gel (Millipore, Billerica, MA) pre-equilibrated with wash buffer containing 25 mM HEPES-NaOH, pH 7.5, 200 mM NaCl and 1% DDM/ 0.1% CHS. The beads were washed with 100 mL of wash buffer containing 25 mM HEPES-NaOH, pH 7.5, 200 mM NaCl, and 1% DDM/ 0.1% CHS and eluted with 4 mL elution buffer containing 25 mM HEPES-NaOH, pH 7.5, 200 mM NaCl, 120 μg/ml of 3xFLAG peptide and 1% DDM/ 0.1% CHS. The purified protein was run on a Superose 6 gel filtration column pre-equilibrated with 0.005% Lauryl Maltose Neopentyl Glycol (LMNG)/0.0005% CHS and the peak fraction was concentrated on 100 K Amicon Ultra-15 concentrators (Millipore, Billerica, MA) to an absorbance at 280 nm of 1.0 to prepare cryo-EM grids.

### SDS-PAGE analysis
An SDS-PAGE analysis was performed to assess the purity of purified proteins. 15 μL of protein sample was supplemented with 5 μL of 4X NuPAGE LDS Sample Buffer (Thermo Scientific). Samples were incubated at 95 °C for 10 min before loading on 4-12% SurePAGE™ Bis-Tris precast gels (Witec AG). Spectra™ Prestained Protein Ladder (Thermo Scientific) (10 to 180 kDa) was also loaded on the gel to run as a size marker. Gels were run in 1X Tris-MOPS SDS running buffer (Witec AG) at 200 V for 30 min, washed briefly in MilliQ water, and stained for 2 h with QuickBlue Protein Stain (LuBioScience GmbH) with shaking. Gels were washed in MilliQ water before imaging on an iBright FL1500 Imaging System (Thermo Scientific).

### Cryo-EM sample preparation and data collection
Purified TRPM4 was incubated with a final concentration of 0.2 mM IBA for TRPM4$_{IBA}$ and NBA for TRPM4$_{NBA}$ for 30 mins at room temperature before freezing cryo-EM grids. Cryo-EM grids were prepared by applying 3 μl of concentrated sample onto 400-mesh R1.2/1.3 UltrAuFoil grids (Quantifoil Micro Tools GmbH), which had rendered hydrophilic by glow discharging at 15 mA for 60 s with a PELCO Easy-Glow device (TED PELLA, INC). The samples were immediately blotted and plunge frozen into liquid ethane using a Vitrobot Mark IV plunge freezer (Thermo Fisher Scientific). Cryo-EM data were collected using the automated data acquisition software EPU (Thermo Fisher Scientific) on a Titan Krios G4 transmission electron microscope (Thermo Fisher Scientific), operating at 300 kV and equipped with a cold-FEG electron source, a SelectrisX energy filter and a Falcon4 direct detection camera. Images were recorded in counting mode at a nominal magnification of 165kx, corresponding to a physical pixel size of 0.726 Å at the sample level. Datasets were collected at a defocus range of 0.8 to 2.5 μm with a total electron dose of 60 e⁻/Å². Image data were saved as Electron Event Recordings (EER).

### Cryo-EM image processing, model building, and refinement
The cryo-EM image processing was performed using cryoSPARC v3.4[35] Patch-based motion correction (cryoSPARC implementation) was used to align the EM movie stacks and apply dose-dependent resolution weighting to recorded movies. CTF estimation was performed using the patch-based option as well. For the data of the HsTRPM4apo, a total of 6429 movies at 0.726 Å per pixel were collected, and 1000 particles were manually picked and used for one round of 2D classification for template creation. Template-based automated particle picking was then used on the recorded image data, which resulted in a

set of 1'719'323 particles at a size of 450 pixels. Two rounds of 2D classification were performed for the initial step of particle cleaning, resulting in 255'053 particles in the first round and 16'308 in the second round. Ab initio and non-uniform refinement yielded one 3D reconstruction with a map at 3.67 Å overall resolution in C4 symmetry (Supplementary Fig. 4).

For the data of the HsTRPM4IBA, a total of 7500 movies at 0.726 Å per pixel were collected. 2D classes from the HsTRPMapo dataset were used for template-based picking. Template-based automated particle picking resulted in a set of 1'149'090 particles at a size of 450 pixels. Two rounds of 2D classification were performed for the initial step of particle cleaning, resulting in 185'741 particles at 400 pixels in the first round and 35'399 particles in the second round. Ab initio and hetero-refinement refinement yielded two 3D reconstructions. One reconstruction representing 57.5% of particles was selected for further non-uniform refinement, resulting in a map at 3.62 Å overall resolution in C4 symmetry. Following symmetry expansion, the overall resolution of the map could be improved to an overall resolution of 3.62 Å (Supplementary Fig. 5).

For the data of the HsTRPM4NBA, a total of 11,851 movies at 0.726 Å per pixel were collected. 2D classes from the HsTRPMapo dataset were used for template-based picking. Template-based automated particle picking resulted in a set of 1'805'182 particles at 400 pixels. Four rounds of 2D classification were performed for the initial step of particle cleaning, resulting in 40'633 particles. Ab initio and hetero-refinement refinement yielded four 3D reconstructions. One reconstruction representing 15'954 particles was selected for further non-uniform refinement, resulting in a map at 4.50 Å overall resolution in C4 symmetry from 15,954 particles (Supplementary Fig. 6).

Cryo-EM data was also collected for DDM/CHS solubilized mouse and human TRPM4. MmTRPM4 data was collected with 5 mM Calcium chloride present in the sample. Two datasets were collected for DDM/CHS solubilized HsTRPM4 with 5 mM Calcium chloride present in one sample and both 5 mM calcium chloride and 0.2 mM NBA present in the other. Details of the data collection are included (Table 2).

Atomic models for HsTRPMapo, HsTRPM4IBA, and HsTRPM4NBA structures were mainly built in Coot 0.9.4[36], using a model PDB id: 6BQV as an initial model. Real-space refinement for all built models was performed using Phenix, version 1.19.2-4158, by applying a general restraints setup[37].

**Cell culture.** TsA-201 cells were cultured with Dulbecco's Modified Eagle's culture Medium DMEM (Gibco, Basel, Switzerland) supplemented with 10% FBS, 0.5% penicillin, and streptomycin (10,000 U/mL) at 37 °C in a 5% $CO_2$ incubator.

**Transfections.** Sixty mm dishes (BD Falcon, Durham, North Carolina, USA) at 80% of confluence were transiently transfected using Xtreme Gene 9™ transfection reagent (Sigma Aldrich Merck, Switzerland) and following the manufacturer's instructions. In brief, 1,000 ng of either empty vector or HsTRPM4 WT or variants of HsTRPM4 (TRPM4 S863A, TRPM4 V901W, TRPM4 V904W, TRPM4 H908A, TRPM4 S924A, TRPM4 K925A, TRPM4 Y1057A, TRPM4 Q1061A, TRPM4 R1064A, TRPM4 R1064G, TRPM4 R1064S or TRPM4 S863A/S924A) was mixed with a solution containing 100 ng of a reporter gene coding for GFP and 1000 ng of empty vector. The expression of GFP was used to evaluate the transfection efficiency procedure, and the empty vector was added to reach a certain amount of total cDNA to ensure an efficient transfection. These final cDNA solutions (2100 ng) were mixed with 210 µL of opti-MEM (Gibco, Basel, Switzerland) and 6.3 µL of Xtreme Gen9™ reagent (ration 1/3). After 30 min at room temperature, the cDNA solutions were applied to the cells. Forty-eight hours post-transfection, the cells were harvested for the western blot experiments.

**Western-blots.** The expression of the human TRPM4 channel was assessed in whole-cell lysates. First, cells were washed with PBS 1X and then lysed for 1 h at 4 °C in lysis buffer (50 mM HEPES pH 7.4, 1.5 mM $MgCl_2$, 150 mM NaCl, 1 mM EGTA pH 8, 10% glycerol, 1% Triton X-100, and Complete® protease inhibitor cocktail (Roche Diagnostics, Mannheim, Germany)). The pellet was discarded after centrifugation at 4 °C, 16,000 g for 15 min. Protein concentrations of each lysate sample were measured in triplicate by Bradford assay and interpolated by a bovine serum albumin (BSA) standard curve. Samples were denatured at 30 °C for 37 min before loading on a gel. Twenty µg of protein for each sample was run at 150 V for 1 hour on 9% polyacrylamide gels. The Turbo Blot dry blot system (Biorad, Hercules, CA, USA) was used to transfer the samples to a nitrocellulose membrane. All membranes were stained with Ponceau as a qualitative check for equivalent total protein loading. Membranes were then rinsed twice with TBS 1X and blocked with 5% BSA in TBS 1X for 1 hour. After this blocking step, the membranes were incubated for 2 h with rabbit anti-human TRPM4 antibody (epitope: $_{1137}$CRDKRESDSERLKRTSQKV$_{1155}$, Pineda, Berlin, Germany) diluted 1:1,000 in TBS 1X + 0.1% tween and mouse anti-$Na^+$/$K^+$ ATPase antibody (Abcam ab 7671) diluted 1:1,000 in TBS 1X + 0.1% tween. The membranes were washed 4 times in TBS 1X + 0.1% tween before incubating with fluorescent secondary antibodies. Secondary antibodies IR Dye 800 CW, anti-rabbit diluted (1:20,000) in TBS 1X + 0.1% tween and IR Dye 700 CW, anti-mouse diluted (1:20,000) in TBS 1X + 0.1% tween (LI-COR Biosciences, Lincoln, NE, USA) were added for 1 h. After 4 washes with TBS 1X + 0.1% tween and 3 washes in TBS 1X, membranes were scanned with the FUSION FX Spectra® Infrared Imaging System (VILBER smart imaging, Marne-la-Vallée, France) to detect fluorescent protein. Subsequent quantitative analysis of protein content was achieved by measuring and comparing band densities (equivalent to fluorescence intensities of the bands) using the *Evolution-Capt software* (VILBER smart imaging, Marne-la-Vallée, France). The background was first subtracted for each band (human TRPM4 and $Na^+$/$K^+$ ATPase), then TRPM4 intensity was divided by the intensity of the $Na^+$/$K^+$ ATPase band (for a given sample) and normalized for comparison.

**Electrophysiology**
**Transfections.** Thirty-five mm dishes (BD Falcon, Durham, North Carolina, USA) at 80% of confluence were transiently transfected using JetPEI™ transfection reagent (Polyplus transfection, Illkirch, France) and following the instructions of the manufacturer. In brief, 500 ng of either empty vector or HsTRPM4 WT or variants of HsTRPM4 (HsTRPM4 T677I, HsTRPM4 S863A, HsTRPM4 V901W, HsTRPM4 V904W, HsTRPM4 H908A, HsTRPM4 S924A, HsTRPM4 K925A, HsTRPM4 Y1057A, HsTRPM4 Q1061A, HsTRPM4 R1064A, HsTRPM4 R1064G, HsTRPM4 R1064S or HsTRPM4 S863A/S924A) was mixed with 200 ng of a reporter gene coding for GFP. Expression of GFP was used to identify transfected cells during patch clamp experiments. Coding DNAs (cDNAs from HsTRPM4 and GFP) were mixed with 4 µl of JetPEI™ and 46 µl of 150 mM NaCl. After 15 min at room temperature, the cDNA solutions were applied to the cells. Twenty-four hours post-transfection, the cells were plated at low density in a new 35 mm dish coated with poly-l-lysine. The cells were patched 24 h post-platting (48 h post-transfection).

**Inside-out patch clamp.** Electrophysiological recordings were performed in the inside-out patch-clamp configuration with patch pipettes (1 and 2 µm tip opening) pulled from 1.5 mm borosilicate glass capillaries (Zeitz-Instruments GmbH, München, Germany) using micropipette puller P 97 (Sutter Instruments, Novato, CA, United States). The tips were polished for 2−4 MΩ pipette resistance in the bath solution. The pipette solution contained 150 mM NaCl, 10 mM HEPES, and 2 mM $CaCl_2$ (pH 7.4 with NaOH). The initial bath solution

with a concentration of 0 μM of free calcium contained 150 mM NaCl, 10 mM HEPES, and 2 mM HEDTA (pH 7.4 with NaOH). After reaching the inside-out configuration, different solutions were perfused at the intracellular side of the membrane patch using a modified rapid solution exchanger (Perfusion Fast-Step SF-77B; Warner Instruments Corp. CT, United States).

<u>For calcium sensitivity experiments.</u> The first solution, applied for 3 to 5 sweeps to reach a stable current, is the bath solution (150 mM NaCl, 10 mM HEPES, and 2 mM HEDTA; pH 7.4 with NaOH) with a concentration of 0 μM of free calcium. Then, to activate TRPM4 channels, a solution (150 mM NaCl, 10 mM HEPES; pH 7.4 with NaOH) containing different concentrations of free calcium (0 μM, 50 μM, 100 μM, 300 μM, 500 μM, 1000 μM, 5000 μM) was applied. The effect of the calcium was quantified when the sodium current of the calcium-activated HsTRPM4 reached stability. Finally, the bath solution with a concentration of 0 μM of free calcium was used to quantify the potential leak current that can occur during such recordings.

<u>For drug dose-response experiments.</u> The first solution, applied for 3 to 5 sweeps to reach the stable current, is the bath solution (150 mM NaCl, 10 mM HEPES, and 2 mM HEDTA; pH 7.4 with NaOH) with a concentration of 0 μM of free calcium. Then, after stabilization of the current, a solution (150 mM NaCl, 10 mM HEPES; pH 7.4 with NaOH) containing a concentration of 300 μM free calcium without compounds was applied to activated TRPM4 channels and recorded the sodium current of the calcium-activated HsTRPM4 until its stabilization. Then, a solution (150 mM NaCl, 10 mM HEPES; pH 7.4 with NaOH) containing a concentration of 300 μM of free calcium and the compound at different concentrations (10 nM, 50 nM, 100 nM, 500 nM, 1000 nM, 5000 nM, 10'000 nM, 50'000 nM) were applied. The drug's effect was quantified when the sodium current of the calcium-activated HsTRPM4 reached stability. Finally, the bath solution with a concentration of 0 μM of free calcium was used to quantify the potential leak current that may occur during such recordings. No investigation and comparison have been made concerning the kinetics of block and washout, and only "stable" currents have been used to calculate the percentage of inhibition.

The sodium current of the calcium-activated HsTRPM4 was recorded with a Multiclamp 700B amplifier (Molecular Devices, Sunnyvale, CA, United States) controlled by Clampex 10 via Digidata 1332 A (Molecular Devices, Sunnyvale, CA, United States). Data were low-pass filtered at 5 kHz and sampled at 10 kHz. Experiments were performed at room temperature (20–25 °C). The holding potential was 0 mV. The stimulation protocol consisted of two pulses totaling 1'000 ms for measuring steady-state currents repeated at 0.2 Hz (1 sweep every 5 s). The first pulse was at −100 mV for 500 ms, and the second was at +100 mV for 500 ms. For analysis, the effect of the compounds on calcium-activated sodium current HsTRPM4 has been calculated by averaging the values of this current during the last 100 ms of the second sweep (at +100 mV) on the trace recorded during the stable phase. Due to the stronger run-down observable at −100 mV (first pulse of the stimulation protocol) for the HsTRPM4 channels leading to, after stabilization of the current, at sometimes almost no sodium current quantifiable at −100 mV of the calcium-activated HsTRPM4 channel (Supplementary Fig. 11b, trace number 3 at −100 mV), the effects of the compounds at −100 mV have not been quantified. Electrophysiology data were exported and analyzed using Prism7.05 GraphPad™ software (GraphPad by Dotmatics, San Diego, CA, USA). Concentration-response curves were fitted using the log(inhibitor) vs. response-variable slope (four parameters) equation (Y=Bottom + (Top-Bottom)/(1 + 10^((LogIC50-X) *HillSlope))) where X corresponds to the log of concentration and Y current recorded in pA. The decadic $\log_{10}$ is used here.

## Data analyses and statistics

Data are represented as means ± SEM. Statistical analyses were performed using Prism7.05 GraphPad™ software (GraphPad by Dotmatics, San Diego, CA, USA). An unpaired Mann-Whitney U test was used to compare two unpaired groups. $p < 0.05$ was considered significant. No muti-group comparison has been performed in this study. Western blots were done in triplicate, and at least 6 cells were used for different doses in electrophysiology experiments. Important information. Due to the limited channels available in our perfusion system (7 channels available), only 5 different concentrations for compounds (NBA or IBA) could be applied to the same cells (the 2 remaining channels were used for the solution with a concentration of 0 μM of free calcium without no compounds to record the background current not related to the activation of HsTRPM4 and the solution with a concentration of 300 μM of free calcium with no compounds to activate the HsTRPM4 channel and record the calcium-activated sodium current). Not all the concentrations (8 in total) for compounds presented in the different figures have been applied to the same cell. This limitation did prohibit the determination of the $IC_{50s}$ per cell; otherwise, the fundamental statistical rule of independence would be violated. Consequently, $IC_{50s}$ were calculated based on the average data point collected for x cells per concentration, as presented in the different figures, and no statistical analysis between the different conditions was performed on those variables. However, a two-way ANOVA with two-factor repeated measures was used to investigate any statistical difference of inhibition, mediated by the compounds, between WT and variant conditions at 1 μM of a compound, a concentration close to the general $IC_{50s}$ of the variants investigated. The percentages of inhibition for all variants, except the TRPM4 T677I, at the concentration of 1 μM of a compound, significantly differed ($p < 0.05$) from the WT and are smaller, suggesting a decrease of inhibition, mediated by the compounds, on these variants of HsTRPM4 channel.

## Data visualization

Gel images were processed and prepared on ImageJ (Version 1.53k). Figures were rendered using PyMOL, (The PyMOL Molecular Graphics System, version 1.8.2.0, Schrödinger, LLC), UCSF Chimera, UCSF ChimeraX[38], and Adobe Illustrator (https://adobe.com/products/illustrator).

## Reporting summary

Further information on research design is available in the Nature Portfolio Reporting Summary linked to this article.

# Data availability

The data that support this study are available from the corresponding authors upon request. The reconstructed maps are available from the EMDB database under access codes EMD-19057 (HsTRPM4apo), EMD-19061 (HsTRPM4IBA), EMD-19069 (HsTRPM4NBA). The atomic models are available in the PDB database, access codes 8RCR (HsTRPM4apo), 8RCU (HsTRPM4IBA) and 8RD9 (HsTRPM4NBA). The micrograph images data are available in the EMPIAR database, access codes EMPIAR-12492 (HsTRPM4apo), EMPIAR-12483 (HsTRPM4IBA) and EMPIAR-12491 (HsTRPM4NBA). A Source Data file is available with this manuscript. Source data are provided with this paper.

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

## Acknowledgements

This work was supported by the Swiss National Science Foundation (SNF Grants CRSII5_177195 and the NCCR TransCure (185544)). We thank Emiko Uchikawa, Sergey Nazarov, and members of the DCI, Lausanne from EPFL, UNIL, and UNIGE for their support. We also thank Kelvin Lau, Florence Pojer, Laurence Durrer, and Soraya Quinche, the PTPSP at EPFL, for their support with protein expression.

## Author contributions

H.S. and H.A. conceptualized the project. B.E. and D.N. performed cryo-EM sample preparation and structure determination with support from the DCI, Lausanne. P.A., A.H., S.G., D.R.K., and M.B. performed biochemistry and patch clamp experiments. C.E.G. and M.L. designed the target molecule IBA and synthesis. C.E.G performed the synthesis of NBA and IBA. Project administration and supervision was done by B.E. and J.S.R.; B.E. wrote the original draft. Manuscript review & editing were done by B.E., D.N., J.S.R., M.L., H.A., and H.S.; H.S. and H.A. acquired funding. All authors read and approved the manuscript.

## Competing interests

The authors declare no competing interests.
