## [Transparent Peer Review file · Nature Communications]

Identification of a Binding Site for Small Molecule Inhibitors Targeting Human TRPM4

Corresponding Author: Professor Henning Stahlberg

Version 0:

Reviewer comments:

Reviewer #1

(Remarks to the Author)

The manuscript by B. Ekundayo et al. presents high-resolution cryo-EM structures of human TRPM4 embedded in native lipid nanodiscs, depicting the channel in unbound, NBA-, and CBA-bound states. The study uncovers a novel binding mode of native cholesterol and identifies potential druggable sites. The work employs state-of-the-art methodologies and represents a significant advancement in comprehending the molecular mechanism and pharmacology of the TRPM4 channel.

There are a few concerns that should be addressed before publication.

Major

1. The authors stated that the TRPM4 channel adopts a more dynamic conformation in native lipid environments, what's the physiological significance? Additionally, further clarification is needed on how native lipids interact with the channel to modify its function. If possible, it would be beneficial for the authors to provide experimental evidence supporting their observations.
2. In Supplementary Figures 3b, 4b, and 5b, the particles seemed to have a low quality in terms of distribution and biochemical homogeneity. This may cause blurred density in the 2D class averages as well as 3D reconstructions, which undermines the conclusion that MHR1/2 may adopt distinct dynamics when the channels is solubilized using LMNG or extracted using SMALP.
3. Main text figures should be improved, and structural analysis should be conducted more thoroughly. Figures (especially Figures 2/3/4) should contain more panels illustrating the side-chain interaction between the drug and the channel to provide a more comprehensive understanding of the binding interactions.
4. The authors' statement regarding the conformational restriction of TRP helix upon NBA/CBA binding requires more explanation and discussion.

Additionally, the schematic representation (Figure 6) of the mechanism needs improvement. Basically, it is hard to interpret from the figure that the binding of NBA/CBA would disrupt the movement of TRP helix, as their binding sites appeared to be located far away from the TRP helix. Furthermore, the conformation of TRP helix, as well as other helices, remained unchanged in all the three states. Moreover, the disruption of the calcium binding site cannot be clearly seen from the figure.

5. The binding site of NBA and CBA is located in the intracellular side of TRPM4. Is there any evidence that showing cell membrane is permeable to the two drugs? Can the antagonistic effects still be observed when testing using whole-cell configuration?

Minor

1. In Figure 3a-3b, no significant conformational change is observed, and the authors also mention that ligand binding induces only subtle changes in the overall structure (line 117). However, in Figure 3c, the radius profile of the pore seems to show significant changes between the apo/NBA-bound and CBA-bound structures. The discrepancies include the total detectable length (distance) of the pore and the location of the minimum radius. These discrepancies need to be addressed and explained.
2. In Figure 4c, the error bars appear to be large. Please specify whether they represent \pm S.D. or \pm S.E.M. Additionally, where applicable, it would be beneficial to provide representative traces of each mutant in the supplementary figures.
3. In Figure 5a, please also clarify whether they represent \pm S.D. or \pm S.E.M.
4. For Figure 5b, please include statistical analysis of the IC50 values.
5. In Supplementary Figures 3b, 4b, and 5b, please label the scale bar on the micrograph.

6. Supplementary Figures 3e, 4e, and 5e. Model-vs-map FSC curves should be provided to evaluate the model quality.
7. Supplementary Figure 8c. Stimulation protocols for the inside-out patch clamp should be clarified on the figure.

Reviewer #2

(Remarks to the Author)

Ekundayo et al provide evidence for the binding site of NBA and CBA compounds to the TRPM4 protein, based on CryoEM structures of TRPM4, in the absence and in the presence of NBA and CBA. TRPM4 is a Ca activated non-selective cation channel that has been implicated in hereditary arrhythmogenic disorders, but also neurodegenerative diseases and cancer. Its value as a drug target in humans is unclear, largely because of a lack of high sensitivity blockers. In that sense this paper promises a true breakthrough in this field, as it provides the first compelling set of data that identifies a binding site on the TRPM4 protein, that could be explored further for compound optimization efforts. However, in its current state, this manuscript does not live up to this promise. Below I have listed some comments that, if addressed properly, will thoroughly improve the persuasiveness of this work.

Major comments:

1. There is a complete lack of description of statistical methods in this paper. In fact, the word statistics is not even mentioned in the complete document. At several occasions p-values are mentioned in the text or in figure legends, but there is no mentioning of which statistical test was used. This is obviously below standard.
2. In the methods section there is no mentioning of the conditions related to the incubation of CBA and NBA to the purified TRPM4 protein. Were CBA and NBA added before or after reconstitution of TRPM4 protein in nano-discs? At what temperature? How long?
3. From the methods section I infer that the TRPM4 protein was purified in the absence of Ca²⁺, and thus all structures capture the protein in the closed state? In that respect line 120 (...in the closed state in all three structures, which is expected as both ligands are inhibitors...) appears somewhat ignorant. Furthermore, I feel that this also poses a conceptual problem to the study. It is not self-evident that a compound will bind to the closed channel in the same way as to the open channel. The conformational change of channel opening could expose another binding site, that is the actual blocking site. Furthermore, open and closed channel blockers have been described amply for other ion channels, such as the NMDA receptor. Thus, the authors should at least exclude at this point that CBA and NBA are exclusive open channel blockers of TRPM4, and that thus one can hypothesise that a compound will bind in the same way to the closed state protein as in the open state.
4. The obtained resolutions of the structures that are presented are somewhat low (above 3.6Å), and especially for the hTRPM4-NBA structure (4.5Å), compared to similar studies in the field. In that respect, it is unclear to me how sure the authors can be of the amino acids that actually interact with the CBA and NBA compounds, and how the binding pocket is composed in Figure 2 and 4. In all of the mutants described in Figure 5, NBA still blocks up to 70% of channel activity. This is hardly a persuasive argument that any of these amino acids is a critical component of the NBA binding to the channel. If any of these residues is critical, block should be largely abolished, right? It is also noteworthy that all mutants lead to a similar reduction on the blocking capacity of NBA. So, all of these residues contribute equally? It would be helpful if the authors could present a negative control here, a mutation that has no effect on blocking efficacy. At this point I'm not convinced at all that the authors point to an actual binding pocket for NBA to TRPM4. To address this, it might be sufficient to create a double mutant to abolish binding of NBA to TRPM4. Actually, as presented now, the data in my view suggest that these residues don't represent the critical binding site, but rather a portion of the protein that blocks access of the compound to an actual binding site. Did the authors consider this?
5. For a complete analysis, the same data as in Figure 5 should also be presented for CBA. This seems obvious to me as they would largely share the binding pocket. I wonder why they are not presented, as this would considerably strengthen their interpretation. The authors miss here the opportunity to explore their hypothesis that the difference in potency of CBA and NBA on human TRPM4 can be found within the binding pocket that they have aligned.
6. In general, the mutants are very scarcely described. There is not a single original trace of these mutant channels in the manuscript, let alone a time course of activity. All functional characterization is done using excised inside out patches and at an intracellular Ca²⁺ concentration of 300µM. It is amply described that this concentration in this configuration does not lead to maximal activation of TRPM4 channels (PMID: 23382873 among others). Furthermore, in an excised patch mode one should expect considerable variation in current as each patch contains a defined number of channels. Therefore, I'm not convinced that Fig4 gives a convincing measure of the relative activity of these mutants compared to the WT channel. This is important, as the authors suggest themselves, as it is conceivable that all of these mutants could interfere with the Ca²⁺ affinity of these modified channels, which appears to be confirmed by a few loss-of-function mutants (V904T, Y1057A, Q1061A and R1064S), which in turn could interfere with the interpretation of the efficacy of NBA/CBA on these channels, because these might also interfere with the Ca²⁺ affinity of the channel, as suggested by the authors. For a convincing analysis, a dose response curve for Ca dependent activation of these mutants should be presented. To bridge the relation between channel expression and activity, it would also be helpful to see a comparison of whole cell currents between WT and mutant channels.
7. Likewise, also the description of block by NBA of these mutants is insufficient. Again, not a single original trace of the mutants with and without blocker. Likewise, no time course of the NBA effect and its washout is presented. It has been amply described before, and co-authors of this study Drs Abriel and Rougier have also shown this (PMID: 24605085), that TRPM4 has a typical 2 phase activation pattern: a transient peak and a more stable plateau phase. To which portion is NBA applied? How long is the blocker applied? How fast is the block of the mutant channel compared to WT (one would expect significant changes of the rate of block when interfering with the critical binding site)? Is the % block calculated based on the steady state effect on channel activity? Or the maximum block within a certain amount of time? How fast is the washout of block from the mutant channel compared to WT? All these elements are necessary to interpret whether the mutations in any

way alter the affinity and/or efficacy of NBA. It is essential that these aspects are better described.

8. It is somewhat disappointing that the authors only illustrate the affinity of 1 mutant channel, S863A, for 1 compound, NBA, in a detailed dose-response analysis. It is also not described why particularly this mutation? A dose response analysis should be the standard for all mutants. A single concentration reporting as in Figure 5a can easily be attributed to change in efficacy without a change in potency (as a measure for affinity) of the compound, not even mentioning the importance of the Hill coefficient. Furthermore, in Fig 5b, they disregard the difference in efficacy (which is illustrated in panel a to calculate an IC50 value, which could lead to misinterpretation of the affinity of the compound for the mutant channel. For a proper interpretation of the change in affinity of CBA and NBA for the TRPM4 channel, a dose response curve should be presented for all mutants. Considering the data as they are presented now, a 10-fold shift in IC50 seems somewhat underwhelming for a residue which should be critical for compound binding? In a comparable study, reference 23, the affinity of TRPM8 for icilin could be completely abolished by comparable single amino acid mutations? Therefore, again, I'm not convinced that any of these residues constitutes a critical binding partner for the compound to the TRPM4 protein.

9. Coming back to my previous remark about statistics, I find it hard to believe in Figure 5 that the V901W bar and K925A bar are not significantly different from the WT condition. Considering the mean value, which is comparable to the S924A bar, and the size of the error bars which are smaller than the S924A bar. But again, no details about statistical approach. Instead, overly detailed p values are mentioned.

10. The authors mention that their proposed mechanism of channel block by CBA and NBA compounds might encompass a change of the Ca²⁺ sensitivity of the channel. However, they don't present any supporting data for this, although all the necessary techniques to do these experiments are present in the current study. So, the obvious question remains, what is then the affinity of the TRPM4 channel for Ca²⁺ in the presence of CBA and NBA? And furthermore, doesn't this mechanism present a conceptual problem for the approach itself, because all structures are determined in Ca²⁺ free condition? Thus, it could be conceivable that in the Ca²⁺ bound state, the current binding site is not accessible? Or that NBA/CBA are expelled from this binding site, when Ca²⁺ binds to the channel?

11. In the discussion section, the authors miss the opportunity to discuss the structural basis for the finding that CBA is human channel specific, as opposed to NBA which inhibits both mouse and human TRPM4.

Minor comments.

1. There are several typos and inconsistencies in the main text and the supplement, some editing is still necessary. Most surprising is the chemical structure of CBA which is shown in Fig 2 and supplementary figure 7. Though the name is cited as 4-chloro-2-[2-(2-46 chloro-phenoxy)-acetyl-amino]-benzoic acid (line 45), the figures depict a 2-iodophenyl ring. Was the iodide variant of CBA used for CryoEM? In that case, could the authors show that the efficacy/potency of this variant to inhibit TRPM4 is the same as for the di-chloride form?

2. The rationale behind the choice of mutants, and especially several variants of the same mutant (R1064A, G and S) should be better described.

3. For the general reader (as once could expect for Nature Communications), it is very difficult to interpret Supplementary table 1. There is no reference in the text to this table and how it should illustrate that the structures that have been obtained are reliable and statistically valid. A bit more explanation on the critical parameters could help.

Version 1:

Reviewer comments:

Reviewer #1

(Remarks to the Author)

The authors had made efforts to avoid overinterpret the results. However, considering this study reports a novel drug binding site, it is of vital importance to discuss the inhibition mechanism or, at least, raising hypothesis. Restricting the movement of TRP helix is generally an acceptable explanation, but should be discussed further.

Additionally, in the current Figure 5, there is almost no new information that was not covered in the previous figures. This is not appropriate for a main text figure. More information is necessary for readers to understand the results, such as diagrams depicting structural differences between TRPM4 in native lipid nanodiscs or detergents, and conformational restriction of TRP helix by NBA/IBA.

Furthermore, in the Supplementary Fig. 8, the local resolution maps are not sufficient for readers to evaluate the ligand density. Please show the ligands together with their surrounding residues simultaneously as isosurface or isomesh, at the same map threshold. At least two viewing angles should be provided.

Reviewer #2

(Remarks to the Author)

The authors have altered the manuscript at several points, and have clearly added additional data, as requested. I would like to thank the authors for those efforts. Yet, I still see a lot of unclarity and confusing data. Overall, I remain at my point that the manuscript in its current state fails to provide conclusive support for its main conclusion. My major comments are listed below:

- The description of statistical methods remains suboptimal. In the methods section it is stated that "An unpaired nonparametric t-test followed by a Mann-Whitney U post-test" was used. I presume the Mann-Whitney test WAS the non-parametric t-test? It is also explicitly stated that no multi-group comparisons were performed. It seems highly unlikely that

this is appropriate. For instance in Figs 3E, Suppl Fig 10A-C, Suppl Fig 13, a non-parametric ANOVA followed by a post-hoc test appears to me highly appropriate. Also, when EC50 and IC50 values are reported, no standard error or deviation is reported, and it is unclear how these values are compared to each other at several occasions in the MS.

- The authors correctly cite the literature that an open pore TRPM4 structure is apparently difficult to obtain even in the Ca bound state. Nevertheless, my original point remains: it is completely unclear from the current set of data whether the binding site for anthranilic compounds which is identified from the CryoEM data is also available in the Ca bound state of the protein. Considering that in HsTRPM4 Ca is coordinated by Glu828, Gln831, Asn865, and Asp868 (PMID: 29217581) and that Ser863 is claimed to be an essential amino acid for NBA binding to TRPM4 (this work), it is very likely that Ca binding to the TRPM4 protein will alter the structure and availability of this binding site. The authors appear to address this by presenting structures of hTRPM4 and mTRPM4 in the presence of Ca, but these structures are derived in DDM/CHS. As the authors point out themselves, in DDM/CHS the binding site is occluded by a cholesterol molecule. The authors further appear to claim that microM Ca present in their SMA buffer would be sufficient to produce a Ca bound structure (Suppl Fig 9a).

However, this was never mentioned in the original structures (and thus, I presume, also not obvious), and also no supporting data for this claim is presented. None of these new structures (Ca bound HsTRPM4 and MmTRPM4) are decently described in the results section. How does the binding pocket for anthranilics look like in the Ca bound condition? Thus, my main point of criticism remains: the authors have identified critical amino acids for the interaction of anthranilic compounds to TRPM4, but I disagree that the available data unequivocally show that this binding pocket is also available in the Ca bound TRPM4 structure and thus, that the presence of anthranilic compounds in this binding pocket actually closes the channel. The possibility remains that these amino acids restrict the access of compounds to another binding site which only becomes accessible in the Ca bound state of the protein. In that respect the importance of the current findings is severely limited.

- The authors try to exclude that the binding site for anthranilic compounds and Ca interfere, by presenting the Ca sensitivity of several TRPM4 variants and of TRPM4 in the presence of NBA. However, this assumption can certainly not be excluded from Suppl Fig 13b. How can one reliably determine the Ca sensitivity of a channel that is 90% blocked? And in neither trace it is clear whether a plateau has been reached that allows to delineate the dose-response curve. And it is also unclear how EC50 values are statistically compared? When comparing the WT trace from panel 13b with the traces of mutants in panel 13a, clear differences in Ca sensitivity appear obvious: for instance Y1057A, R1064S, and S863A/S924A have an EC50 value which is almost 50% lower compared to WT, indeed indicating that the suggested binding site of anthranilic compounds and Ca interacts? Thus, I remain at my point that a CryoEM structure in the presence of Ca and anthranilics is essential for a proper conclusion whether the anthranilic binding site which is suggested here is also present in the Ca bound protein, or that other conclusive data should be presented that Ca binding to TRPM4 does not interfere with the anthranilic binding site.

- Why exactly the Ca dependence of the specific variants in Fig13 is determined is also not clear at all. When correlating data with Suppl Fig 12, Y1057A, Q1061A and R1064S show hardly any activity in the patches. Again, how can one reliably determine the Ca dependence of these loss-of-function variants. On the other hand, S924A appears from Fig 12 a gain-of-function variant, but unfortunately Ca²⁺ dependence of this variant is not determined?

- I don't understand the rationale behind the use of IBA, instead of CBA that was extensively characterized by the (co-)authors? Are properties of IBA and CBA the same? IBA blocks the channel (though that is not so clear from Fig 2B, see my point below), but is it also human specific, as CBA? If it binds in the same binding pocket as NBA, how do the authors then explain that the one is HsTRPM4 specific and the other is not, or is this not the case? This is a very important topic, which deserves more attention. Furthermore, the authors assert that the interactions between the anthranilic acid moiety and residues of TRPM4 elucidate the chemical foundation underlying the specific binding of anthranilic acid derivatives to TRPM4. However, as illustrated in Figure 3, the orientation of the anthranilic moiety differs fundamentally, with the carboxyl of NBA facing S3 and S4, and of IBA facing the S4-S5 linker. The anthranilic moiety serves as a core structure for this class of TRPM4 antagonists, and is essential for their activity (PMID: 29579323). It is, at the very least, highly surprising that there exists such a significant difference in orientation, and this should be addressed by the authors.

- Coming back to Fig2B: I infer from the methods section that block of the channel throughout the paper is calculated at +100mV. Yet, at least for IBA (from Fig2B) it appears that at -100mV the effect is a lot smaller, or even absent? So IBA does not block inward currents? These data suggest that block of hTRPM4 by IBA is voltage-dependent, is that the case? And is that also the case for NBA? From the traces it appears not, which would suggest that NBA and IBA have a completely different mode of action? Does it then make sense that they bind the same pocket? The authors should address this.

Minor:

- It is somewhat disappointing that no whole cell currents of TRPM4 variants are reported. Suppl Fig 10 suggests strong differences in activity (gain- and loss-of-function) of the different channel variants, but it is unclear whether this is the result of the variation in the number of channels in the excised patch. The variation in the western blot data is so large that hardly a conclusion is possible.

- Overall, the text contains numerous typo's and needs editing. Also, it appears to me that there should be more data presented in the main figures. I suggest transferring data from the supplementary figures to the main figures.

Version 2:

Reviewer comments:

Reviewer #1

(Remarks to the Author)

All my concerns have been addressed.

Reviewer #2

(Remarks to the Author)

The authors have not fundamentally addressed the issues that I brought up in my previous review. It is clear that this is an extensive piece of work, but the issue of an interrelation between the IBA/NBA and the Ca binding site of the protein and the way that it is being treated in this MS remains an important issue in my view that limits the decisiveness of this work. This should at least be critically discussed in the discussion section of the paper.

I specifically disagree with the statistical evaluation of EC50 and IC50 values, which is apparently impossible? What is against determining the EC50 value for each cell, and comparing them between WT and mutants? I advise that a statistical expert goes through the methodology of the paper.

Furthermore, about my comment on the orientation of NBA and IBA in the binding pocket of the original MS, the authors now comment that "We have reanalyzed the fitting of the ligands into the density and have better fitted them (sic), and they are now more similar. We have changed this in Figure 3." I'm very surprised by the looseness of this adaptation. Was the previous fitting not well performed? Which parameters determine whether the compounds are "better fitted"? To me this indicates that there is considerable freedom of the compounds in this binding pocket and that it is unclear how the authors can determine the "correct" orientation of IBA or NBA in this binding pocket. This uncertainty should be highlighted in the manuscript, and is worth to be discussed.

I also find the treatment of TRPM4 currents at -100mV as unreliable ("...we never quantify it, and we think that we must not draw any conclusion on this part") quite disturbing. Many papers have analysed in the past both inward and outward currents through TRPM4. Negative potentials are of course the physiological range in which the channel is active, and the fact that currents in the presence of the blocker are very variable in this voltage range might well indicate that the block of TRPM4 by anthranilic compounds is voltage dependent. At least, this way of analysing and the reason for it should be clearly highlighted in the manuscript, so that it is clear to the reader why this is done in this way.

Finally, during the review process an important manuscript in the same field was published (Hu et al., Nature, PMID: 38750366). They highlight the importance of temperature for the confirmation of the TRPM4 protein. The implications of this work for the current manuscript should be addressed in the discussion.

Version 3:

Reviewer comments:

Reviewer #2

(Remarks to the Author)

It should be more clear from the legend and make-up of Figure 4a and Suppl Fig 9 (top panel) that the authors cannot exclude that the density observed in the Ca binding site is not actually a Ca ion (as correctly outlined in the results section, lines 130-144). Now it is clearly depicted as a Ca ion (green ball, with Ca annotated). I suggest another color (compared to the DDM:CHS condition where more confidence exists that Ca) and removing Ca as annotation in the figure. Actually, in the method section on protein solubilisation there is no mentioning on how much Ca was added in the DDM:CHS condition, nor is it obvious how much 'endogenous Ca' could be expected in the SMA condition. One could assume that in the absence of any Ca buffer and using purified components that lower microM concentrations of Ca are still present in the 'no Ca added condition', but as is obvious to the authors, this is hardly sufficient to activate TRPM4, and thus it is actually highly unlikely that Ca is present in the Ca binding pocket in the SMA-no added Ca condition. I'm satisfied with how this is handled in the text, but the figure legend and the figure itself should reflect this interpretation.

I have no other questions or remarks.

Responses to Reviewers.

We thank both reviewers for the detailed reviews/ comments of our manuscript. We have addressed the concerns raised, which have significantly improved the quality of the manuscript. We thank the reviewers for this. Our responses to reviewers are given. We would also like to mention that we have used the 3-iodophenyl version of CBA in our experiments, which we named IBA (IUPAC systematic name: 4-chloro-2-(2-(3-iodophenoxy)acetamido)benzoic acid). We apologise for the mislabelling and confusion caused. This has now been corrected in the whole manuscript, materials & methods section, and supplementary information.

Reviewer #1 (Remarks to the Author):

The manuscript by B. Ekundayo et al. presents high-resolution cryo-EM structures of human TRPM4 embedded in native lipid nanodiscs, depicting the channel in unbound, NBA-, and CBA-bound states. The study uncovers a novel binding mode of native cholesterol and identifies potential druggable sites. The work employs state-of-the-art methodologies and represents a significant advancement in comprehending the molecular mechanism and pharmacology of the TRPM4 channel.

There are a few concerns that should be addressed before publication.

Major

1. The authors stated that the TRPM4 channel adopts a more dynamic conformation in native lipid environments, what's the physiological significance? Additionally, further clarification is needed on how native lipids interact with the channel to modify its function. If possible, it would be beneficial for the authors to provide experimental evidence supporting their observations.

We thank the reviewer for this comment. We also put into consideration Comment 2 below and have removed the claims on dynamics as, indeed the biochemical stability of the sample is less in the SMALP solubilised sample. We now highlight a crucial point of this study: since we had less density for annular lipids in the TMD region, we can now clearly identify the presence of our ligands in the CryoEM maps see **lines 114-130**. Although the sample had less biochemical homogeneity than solubilised with DDM/CHS, the ligand binding site, which is occluded by a CHS molecule in the DDM/CHS solubilised samples, is now available for ligand binding in our SMALP sample as shown in Supplementary Figure 2. We have also included **a new Supplementary Figure 3**, where we provide three additional cryoEM maps we have produced from this study of DMM/CHS solubilised human and mouse TRPM4. The ligand binding site is seen to be occluded by a CHS molecule also in the map of Human and Mouse TRPM4 when the inhibitor (NBA) is added.

2. In Supplementary Figures 3b, 4b, and 5b, the particles seemed to have a low quality in terms of distribution and biochemical homogeneity. This may cause blurred density in the 2D class averages as well as 3D reconstructions, which undermines the conclusion that MHR1/2 may adopt distinct dynamics when the channels is solubilized using LMNG or extracted using SMALP.

Our response to comment 1 above also deals with Comment 2.

3. Main text figures should be improved, and structural analysis should be conducted more thoroughly. Figures (especially Figures 2/3/4) should contain more panels illustrating the side-chain interaction between the drug and the channel to provide a more comprehensive understanding of the binding interactions.

We have made extensive changes to the main text **Figures 2, 3 and 4** to illustrate side-chain interactions, and electrophysiology data is also shown.

4. The authors' statement regarding the conformational restriction of TRP helix upon NBA/CBA binding requires more explanation and discussion. Additionally, the schematic representation (Figure 6) of the mechanism needs improvement. Basically, it is hard to interpret from the figure that the binding of NBA/CBA would disrupt the movement of TRP helix, as their binding sites appeared to be located far away from the TRP helix. Furthermore, the conformation of TRP helix, as well as other helices, remained unchanged in all the three states. Moreover, the disruption of the calcium-binding site cannot be clearly seen from the figure.

We have now removed our claim concerning the mechanism of inhibition and focused more on the identification of the ligand-binding site. A mechanistic description of inhibition would require an open-state structure, which we are not able to obtain in our study. Therefore, we focus the manuscript on identifying and validating the ligand binding site for further drug development. A **new Figure 5 (before Figure 6)** highlights the ligand binding with the mechanistic details removed. We have also included a new supplementary Figure 9, where we show the cryo-EM density for Calcium in SMALP and DMM/CHS solubilized TRPM4 samples generated in this study. This is also reflected in the new title: **Identification of a Binding Site for Small Molecule Inhibitors Targeting Human TRPM4**

5. The binding site of NBA and CBA is located in the intracellular side of TRPM4. Is there any evidence that showing cell membrane is permeable to the two drugs? Can the antagonistic effects still be observed when testing using whole-cell configuration?

Thanks to the reviewer for this comment. As previously reported, the antagonistic effect of the compound NBA is observed when using the whole configuration approaches in patch clamp experiments with LNCaP cells (PMID 29579323) and HCT116 cells (PMID 34771564). Overall, these data suggest the cell membrane is permeable to these drugs. Moreover, the theoretical lipophilicity of these drugs (clog P) has been calculated using the algorithms on Swiss ADME (www.swissadme.ch) from the Swiss Institute of Bioinformatics. At pH = 7.4, compounds NBA and IBA (both protonated and deprotonated forms) tend to be lipophilic. The NBA is slightly more lipophilic than IBA (consensus clog P values protonated/deprotonated: NBA: 3.59/3.32; IBA: 3.25/3.05). These values indicate that these drugs likely go through the plasma membrane of the cells. This is mentioned in **lines 975-992** in the materials and methods section of the paper.

Minor

1. In Figure 3a-3b, no significant conformational change is observed, and the authors also mention that ligand binding induces only subtle changes in the overall structure (line 117). However, in Figure 3c, the radius profile of the pore seems to show significant changes

between the apo/NBA-bound and CBA-bound structures. The discrepancies include the total detectable length (distance) of the pore and the location of the minimum radius. These discrepancies need to be addressed and explained.

We thank the reviewer for highlighting this. However, the differences in the radius profile of the pore are only due to rotamer differences in side chains of a few residues lining the pore observed when model building for fitting in Cryo-EM maps. Notably, rotamer differences with the last residue of the pore S1044 lead to an increase in the pore length. Although these change the profile, the overall profile shows a closed state of the pore, similar to other structures. As pore profiles could be sensitive to $<1\text{\AA}$ changes in side chain conformations, further claims about the subtle changes observed are not justified at a local resolution of 3\AA in this region of our Cryo-EM map. We have also included a statement concerning this in **lines 173-176**.

2. In Figure 4c, the error bars appear to be large. Please specify whether they represent $\pm\text{S.D.}$ or $\pm\text{S.E.M.}$ Additionally, where applicable, it would be beneficial to provide representative traces of each mutant in the supplementary figures.

All statistics and error bars have been reevaluated as requested. The new error bars represent the s.e.m. As requested, traces of each single mutant have been added in **supplementary Figure 12**

3. In Figure 5a, please also clarify whether they represent $\pm\text{S.D.}$ or $\pm\text{S.E.M.}$

This information has been added to the statistical section as requested. The new error bars represent the s.e.m.

4. For Figure 5b, please include statistical analysis of the IC_{50} values.

This information has been added to the new version of this manuscript as requested.

5. In Supplementary Figures 3b, 4b, and 5b, please label the scale bar on the micrograph.

We have included labelled scale bars on the micrographs

6. Supplementary Figures 3e, 4e, and 5e. Model-vs-map FSC curves should be provided to evaluate the model quality.

We have included these in the figures which are now **Supplementary Figures 4, 5 and 6**.

7. Supplementary Figure 8c. Stimulation protocols for the inside-out patch clamp should be clarified on the figure.

For clarity purposes, Supplementary figure 8C has been removed from the new manuscript. The new **Supplementary figures 11 and 12** now contain the inside out patch clamp data. However, the material and methods section still presents the protocol used. Moreover, the protocol has been used in the different figures as requested.

Reviewer #2 (Remarks to the Author):

Ekundayo et al provide evidence for the binding site of NBA and CBA compounds to the TRPM4 protein, based on CryoEM structures of TRPM4, in the absence and in the presence of NBA and CBA. TRPM4 is a Ca activated non-selective cation channel that has been implicated in hereditary arrhythmogenic disorders, but also neurodegenerative diseases and cancer. Its value as a drug target in humans is unclear, largely because of a lack of high sensitivity blockers. In that sense this paper promises a true breakthrough in this field, as it provides the first compelling set of data that identifies a binding site on the TRPM4 protein, that could be explored further for compound optimization efforts. However, in its current state, this manuscript does not live up to this promise. Below I have listed some comments that, if addressed properly, will thoroughly improve the persuasiveness of this work.

Major comments:

1. There is a complete lack of description of statistical methods in this paper. In fact, the word statistics is not even mentioned in the complete document. At several occasions p-values are mentioned in the text or in figure legends, but there is no mentioning of which statistical test was used. This is obviously below standard.

Thanks to the reviewer for this comment; we apologize for this mistake. A statistical section has been added in the material and methods of the new version of the manuscript **lines 1444-1450**. The entire statistic has been performed de novo in the latest version of the manuscript.

2. In the methods section there is no mentioning of the conditions related to the incubation of CBA and NBA to the purified TRPM4 protein. Were CBA and NBA added before or after reconstitution of TRPM4 protein in nano-discs? At what temperature? How long?

This information has now been included in lines 1068-1069 in the materials and methods section of the new manuscript.

3. From the methods section I infer that the TRPM4 protein was purified in the absence of Ca^{2+} , and thus all structures capture the protein in the closed state? In that respect line 120 (...in the closed state in all three structures, which is expected as both ligands are inhibitors...) appears somewhat ignorant. Furthermore, I feel that this also poses a conceptual problem to the study. It is not self-evident that a compound will bind to the closed channel in the same way as to the open channel. The conformational change of channel opening could expose another binding site, that is the actual blocking site. Furthermore, open and closed channel blockers have been described amply for other ion channels, such as the NMDA receptor. Thus, the authors should at least exclude at this point that CBA and NBA are exclusive open channel blockers of TRPM4, and that thus one can hypothesise that a compound will bind in the same way to the closed state protein as in the open state.

We thank the reviewer for highlighting this. We also want to highlight that there is no published structure of Trpm4 in the open state, even when calcium ions are present in the structure (PMID: 29217581). Thus far, the addition of Ca^{2+} ions is insufficient to open the channel in detergent micelle or nanodiscs. We have also observed density in our structure into which Ca^{2+} ions can be fitted, as shown in **Supplementary Figure 9** when compared to a previously determined structure in nanodisc in which Ca^{2+} ions are present (PMID: 29217581). We also have included 2 structures of DDM/CHS structures of human and mouse Trpm4 with Ca^{2+} ions bound showing a similar density for Ca^{2+} present (**Supplementary Figure 9**). Although additional Ca^{2+} ions were not added to the sample before freezing for CryoEM, we also did not add EDTA. Therefore, it is possible that the

endogenous Ca^{2+} ions is bound. Also, the structure we determined is similar to the previously determined structures of Trpm4 with Ca^{2+} ions present.

Although we cannot rule out a difference in the binding site in the open and closed state, we have performed extensive electrophysiology experiments with ligand binding site mutant variants of Trpm4. These data also strongly support and validate that the binding site that we have proposed in this work is the inhibitor binding site. Our goal was to identify the inhibitor-binding site which our electrophysiology data support as mutations in the binding alleviate inhibition and this effect is increased in a double mutation in the binding site as shown in **Figure 4**.

4. The obtained resolutions of the structures that are presented are somewhat low (above 3.6Å), and especially for the hTRPM4-NBA structure (4.5Å), compared to similar studies in the field. In that respect, it is unclear to me how sure the authors can be of the amino acids that actually interact with the CBA and NBA compounds, and how the binding pocket is composed in Figure 2 and 4. In all of the mutants described in Figure 5, NBA still blocks up to 70% of channel activity. This is hardly a persuasive argument that any of these amino acids is a critical component of the NBA binding to the channel. If any of these residues is critical, block should be largely abolished, right? It is also noteworthy that all mutants lead to a similar reduction on the blocking capacity of NBA. So, all of these residues contribute equally?

Thank you for this comment. Although the resolutions of the structure are lower than 3.6Å the local resolutions in the regions of the Transmembrane domain (TMD) and ligand binding site are relatively better for interpretation of the amino acids interactions in the binding site (see Supplementary Figures 4, 5, 6 and 8). It is also important to note that the resolution reported in our study is a significant improvement from previous attempts to obtain a structure of Trpm4 in the native nanodiscs ([https://www.cell.com/chem/pdfExtended/S2451-9294\(20\)30414-9](https://www.cell.com/chem/pdfExtended/S2451-9294(20)30414-9)). We also show that we can obtain higher resolution structures of Trpm4 in detergent micelles however the native nanodisc structures were necessary for us to identify the ligand binding site.

However, we believe that not only one amino acid in this binding pocket is critical. Based on the data, we propose that many amino acids are necessary for interacting with the compound, which may explain the slight effect observed by mutating only one of them. As suggested by the reviewer, a double mutant (hTRPM4 p.S863A/S924A) has been generated and studied. As presented in the new version of the manuscript, this double mutant, compared to a single mutant, further decreases the inhibitory effect mediated by the NBA compounds but does not abolish it, strongly supporting the model that all these residues contribute equally.

It would be helpful if the authors could present a negative control here, a mutation that has no effect on blocking efficacy.

A negative control (HsTRPM4 T677I) has been added in the new version, as suggested by the reviewer in **Figure 3f and g**. It confirms that a mutation localized outside the predicted binding pocket of the compound does not influence the inhibitory effect mediated by NBA.

At this point I'm not convinced at all that the authors point to an actual binding pocket for NBA to TRPM4. To address this, it might be sufficient to create a double mutant to abolish binding of NBA to TRPM4.

Thanks to the reviewer for this comment. As already mentioned, a double mutant has been added, as requested as shown in **Figure 4**. As presented in the new version of the manuscript, this double mutant, compared to a single mutant, further decreases the inhibitory effect mediated by the NBA compounds, suggesting that the amino acids investigated in this study are part of the binding pocket for the NBA compound.

Actually, as presented now, the data in my view suggest that these residues don't represent the critical binding site, but rather a portion of the protein that blocks access of the compound to an actual binding site. Did the authors consider this?

It is an exciting point. However, if this model is correct, we may expect to improve the accessibility to the drug for its "real" binding site by mutating those amino acids shown to interact with the drug. These mutations may enhance the inhibitory effect mediated by the NBA, which is not experimentally observed. Based on the data, we are still convinced that the amino acids identified are part of the actual binding pocket for the NBA.

5. For a complete analysis, the same data as in Figure 5 should also be presented for CBA. This seems obvious to me as they would largely share the binding pocket. I wonder why they are not presented, as this would considerably strengthen their interpretation. The authors miss here the opportunity to explore their hypothesis that the difference in potency of CBA and NBA on human TRPM4 can be found within the binding pocket that they have aligned.

Thanks to the reviewer for this suggestion. As mentioned in the manuscript, the functional experiments were done with NBA since it is the more potent inhibitor. The decision to use only one compound (NBA) for the proof of concept has also been made based on time-consuming experiments related to the recordings. Moreover, a substantial amount of data was added in the new version of the manuscript investigating the NBA and its binding pocket in the timeframe of three months, interfering with the possibility of doing similar experiments using the IBA compound.

6. In general, the mutants are very scarcely described. There is no original trace of these mutant channels in the manuscript, let alone a time course of activity. All functional characterization is done using excised inside out patches and at an intracellular Ca^{2+} concentration of $300\mu\text{M}$. It is amply described that this concentration in this configuration does not lead to maximal activation of TRPM4 channels (PMID: 23382873 among others). Furthermore, in an excised patch mode one should expect considerable variation in current as each patch contains a defined number of channels. Therefore, I'm not convinced that Fig4 gives a convincing measure of the relative activity of these mutants compared to the WT channel. This is important, as the authors suggest themselves, as it is conceivable that all of these mutants could interfere with the Ca^{2+} affinity of these modified channels, which appears to be confirmed by a few loss-of-function mutants (V904T, Y1057A, Q1061A and R1064S), which in turn could interfere with the interpretation of the efficacy of NBA/CBA on these channels, because these might also interfere with the Ca^{2+} affinity of the channel, as suggested by the authors. For a convincing analysis, a dose response curve for Ca

dependent activation of these mutants should be presented. To bridge the relation between channel expression and activity, it would also be helpful to see a comparison of whole cell currents between WT and mutant channels.

Thanks to the reviewer for this point. Traces and time courses have been added as requested as shown in **Figures 2, 4 and Supplementary figures 11, 12 and 14**. As highlighted by the reviewer with the publication mentioned, the 300 μM of free calcium is probably insufficient to “fully” activate the TRPM4 channel. Moreover, the fact that the inside-out approach may lead to variability due to pipette size fluctuation is correct (although we tried to keep similarly sized pipettes during all the experiments). However, the whole-cell approach is also imperfect due to other limitations (e.g., transfection efficiency issues from experiment to experiment or stability of the recording). To circumvent such a problem, intrinsic to the methods, the increase in the number of cells recorded is generally an approach used. Nevertheless, to avoid misinterpretation of the data, we performed, as suggested by the reviewer, the entire dose-effect curves (calcium sensitivity) for a few mutants, leading some of them to a loss of function as shown in **Supplementary Figure 13a**. As now presented in the new version, the calcium sensitivity of these mutants is not reduced compared to the WT TRPM4, suggesting that the decrease of the currents observed may not be linked to a reduction in calcium sensitivity. Based on this information, it can be speculated that the decrease in protein expression or localization may explain this loss of function observed (as shown, for example, on the western blot for the mutant TRPM4 Q1061A) **as shown in Supplementary Figure 1**.

In addition, calcium sensitivity curves have been performed with the wild-type channel in the presence or absence of NBA to investigate the potential alteration of this sensitivity mediated by the presence of NBA compounds. As present in the new version of this manuscript, no difference has been observed for this calcium sensitivity in the presence or absence of NBA as shown in **Supplementary Figure 13b**.

7. Likewise, also the description of block by NBA of these mutants is insufficient. Again, not a single original trace of the mutants with and without blocker. Likewise, no time course of the NBA effect and its washout is presented. It has been amply described before, and co-authors of this study Drs Abriel and Rougier have also shown this (PMID: 24605085), that TRPM4 has a typical 2 phase activation pattern: a transient peak and a more stable plateau phase. To which portion is NBA applied? How long is the blocker applied? How fast is the block of the mutant channel compared to WT (one would expect significant changes of the rate of block when interfering with the critical binding site)? Is the % block calculated based on the steady state effect on channel activity? Or the maximum block within a certain amount of time? How fast is the washout of block from the mutant channel compared to WT? All these elements are necessary to interpret whether the mutations in any way alter the affinity and/or efficacy of NBA. It is essential that these aspects are better described.

Thanks to the reviewer for this comment. As requested, more information/traces have been added in the new version. As mentioned, no whole-cell approaches have been used for this study due to the limitation concerning the current stability (two phases for which the molecular mechanism is still not clarified) compared to the inside-out approaches providing a stable calcium-activated current more rapidly. In general, 15-20 minutes is required to reach

a steady current using the whole-cell approach compared to 4-5 minutes for the inside-out approach (**Supplementary Figure 11**).

In terms of procedure: After obtaining the inside-out configuration, the protocol consists of perfusing the intracellular part of the membrane with a solution without calcium-free during 3-5 sweeps and then with solutions containing 300 μ M calcium-free to activate TRPM4 channels. After reaching stability, the drug in solution containing the same amount of calcium-free was applied to investigate the inhibitory efficiency of this latest. After gaining stability in the presence of the drug, a solution without compound and calcium-free is perfused to quantify the potential leak current occurring during the recording. This information has been clarified in the new version of the manuscript.

In brief, no investigation and comparison have been made concerning the kinetic of block and washout and only "stable" currents have been used to calculate the percentage of inhibition. Nevertheless, it is correct that a deep analysis of the different kinetic parameters will be fascinating; however, it was not our priority in this publication in which we present the binding pocket of TRPM4 for the NBA compounds for the first time using CryoEM. Nevertheless, preliminary data suggest that the NBA is washable and the kinetic inhibition between the WT and one mutant (TRPM4 S933A) are similar. However, because not all the mutants have been investigated, we prefer not to mention those data in the new version to avoid over-interpretation.

8. It is somewhat disappointing that the authors only illustrate the affinity of 1 mutant channel, S863A, for 1 compound, NBA, in a detailed dose-response analysis. It is also not described why particularly this mutation? A dose response analysis should be the standard for all mutants. A single concentration reporting as in Figure 5a can easily be attributed to change in efficacy without a change in potency (as a measure for affinity) of the compound, not even mentioning the importance of the Hill coefficient. Furthermore, in Fig 5b, they disregard the difference in efficacy (which is illustrated in panel a to calculate an IC₅₀ value, which could lead to misinterpretation of the affinity of the compound for the mutant channel. For a proper interpretation of the change in affinity of CBA and NBA for the TRPM4 channel, a dose response curve should be presented for all mutants.

Thanks for this valid point. This mutation was used based on a similar expression to the wild-type and decreased NBA efficiency. However, based on other suggestions/comments from the reviewer, a complete set of dose-effect curves has been performed with more TRPM4 mutants (TRPM4 S863A, TRPM4 V901W, TRPM4 H908A, TRPM4 S924A, TRPM4 R1064A, TRPM4 R1064G, and TRPM4 S863A/S924A) and are now presented in the new version of this manuscript as shown in **Supplementary Figure 13a**. These mutants have been selected to cover as much all possible effects mediated by those mutations compared to the wild-type channels (e.g., loss of expression (TRPM4 S863A), loss of function (TRPM4 V901W, TRPM4 H908A, TRPM4 R1064A, and TRPM4 S863A/S924A), and no difference (TRPM4 S924A, and TRPM4 R1064G)). Compared to the wild-type, 6 of 7 mutants showed a profound augmentation of the IC₅₀ for NBA, suggesting a decrease in inhibition as shown in **Figure 3e**. This effect is more pronounced for the double mutant compared to the wild-type, suggesting that the double mutant diminishes the inhibitory effect mediated by NBA more drastically than the single mutant.

The Hill coefficient has not been presented mainly because the number of cells is not powerful enough to interpret these data reliably. The fit used for the analysis is based on a variable slope to avoid too much constraint (the slope is not fixed to 1, for example, as it is in

many cases). However, such an approach requires a lot of cells per condition to interpret the Hill coefficient information provided by the software reliably. Based on this limitation, we do not present the Hill coefficient to avoid misinterpreting these values.

Considering the data as they are presented now, a 10-fold shift in IC₅₀ seems somewhat underwhelming for a residue which should be critical for compound binding? In a comparable study, reference 23, the affinity of TRPM8 for icilin could be completely abolished by comparable single amino acid mutations? Therefore, again, I'm not convinced that any of these residues constitutes a critical binding partner for the compound to the TRPM4 protein.

As mentioned in our answers to the reviewer, we are firmly convinced that all these amino acids investigated in this study are necessary to bind the NBA compound. Although some interaction between the compounds and the protein may be highly dependent on one amino acid, as presented in reference 23, other studies, for example, the one investigating the calcium sensitivity of TRPM4 and TRPM5, show similar effects to the one presented in this study (a single mutation of the calcium-binding site does not abolish the sensitivity to the calcium of the channel), suggesting that no consensus exist (PMID 31022885).

9. Coming back to my previous remark about statistics, I find it hard to believe in Figure 5 that the V901W bar and K925A bar are not significantly different from the WT condition. Considering the mean value, which is comparable to the S924A bar, and the size of the error bars which are smaller than the S924A bar. But again, no details about statistical approach. Instead, overly detailed p values are mentioned.

Thanks to the reviewer for this comment. We apologize for this mistake. All the statistics have been performed de novo in the new version.

10. The authors mention that their proposed mechanism of channel block by CBA and NBA compounds might encompass a change of the Ca²⁺ sensitivity of the channel. However, they don't present any supporting data for this, although all the necessary techniques to do these experiments are present in the current study. So, the obvious question remains, what is then the affinity of the TRPM4 channel for Ca²⁺ in the presence of CBA and NBA? And furthermore, doesn't this mechanism present a conceptual problem for the approach itself, because all structures are determined in Ca²⁺ free condition? Thus, it could be conceivable that in the Ca²⁺ bound state, the current binding site is not accessible? Or that NBA/CBA are expelled from this binding site, when Ca²⁺ binds to the channel?

We thank the reviewer for this critical comment. As mentioned above, a calcium sensitivity experiment using a wild-type HsTRPM4 construct has been performed in the presence or absence of NBA (200 nM) to investigate the potential alteration of this sensitivity due to the binding of the drug. The data reported in the new version of the manuscript suggest no drastic difference for this parameter (EC₅₀) as shown in **Supplementary Figure 13b**. We have also removed the mechanism we proposed from the manuscript and focused more on the drug-binding site.

11. In the discussion section, the authors miss the opportunity to discuss the structural basis for the finding that CBA is human channel specific, as opposed to NBA which inhibits both mouse and human TRPM4.

We apologise for the discrepancy but in this study we have not used CBA for structure determination but IBA which is a different derivative. WE show that both IBA and CBA have comparable potency, as shown in **Figure 2a -c**.

Minor comments.

1. There are several typos and inconsistencies in the main text and the supplement, some editing is still necessary. Most surprising is the chemical structure of CBA which is shown in Fig 2 and supplementary figure 7. Though the name is cited as 4-chloro-2-[2-(2-46 chloro-phenoxy)-acetylamino]-benzoic acid (line 45), the figures depict a 2-iodophenyl ring. Was the iodide variant of CBA used for CryoEM? In that case, could the authors show that the efficacy/potency of this variant to inhibit TRPM4 is the same as for the di-chloride form?

The reviewer is correct. We have used the 3-iodophenyl version of CBA in our experiments, which we named IBA (IUPAC systematic name: 4-chloro-2-(2-(3-iodophenoxy)acetamido)benzoic acid). We apologize for the mislabelling and confusion caused. This has now been corrected in the whole manuscript, materials & methods section, and supplementary information.

2. The rationale behind the choice of mutants, and especially several variants of the same mutant (R1064A, G and S) should be better described.

The choice of mutants was based on interactions with the ligand. Several mutations to the R1064 residue was made as it is contained in the ClinVar Miner database (https://clinvarminer.genetics.utah.edu/submissions-by-variant/NM_017636.4%28TRPM4%29%3Ac.3190C%3EG%20%28p.Arg1064Gly%29) as possibly being disease relevant which is why we included these in our study just for our own curiosity to see if there would be an effect.

3. For the general reader (as once could expect for Nature Communications), it is very difficult to interpret Supplementary table 1. There is no reference in the text to this table and how it should illustrate that the structures that have been obtained are reliable and statistically valid. A bit more explanation on the critical parameters could help.

We apologise for the omission. We have now added the reference in text in line 112. The table contains the standard metrics for data collection and quality for Cryo-EM. We have also described this in the material and methods section under CryoEM image processing, model building and refinement. The quality of the model to the map in different parts of the structure can be better visualized in Supplementary Figure 7 for the general reader.

REVIEWER COMMENTS

We would like to thank the reviewers for their comments, which have helped us to substantially improve the clarity and quality of the manuscript. Our responses to the reviewers are attached below:

Reviewer #1 (Remarks to the Author):

The authors had made efforts to avoid overinterpret the results. However, considering this study reports a novel drug binding site, it is of vital importance to discuss the inhibition mechanism or, at least, raising hypothesis. Restricting the movement of TRP helix is generally an acceptable explanation, but should be discussed further.

We thank the reviewer and agree that this is an important point. Therefore, we have included our hypothesis for the inhibition mechanism in Figure 6, also described in lines 227-237.

Additionally, in the current Figure 5, there is almost no new information that was not covered in the previous figures. This is not appropriate for a main text figure. More information is necessary for readers to understand the results, such as diagrams depicting structural differences between TRPM4 in native lipid nanodiscs or detergents, and conformational restriction of TRP helix by NBA/IBA.

We thank the reviewer for this point. We have revised the former Figure 5 to Figure 6 to show possible conformational restrictions of the TRP helix by NBA/IBA. Supplementary Figures 2a and b show differences in the lipid environment between native lipid nanodisc and detergents.

Furthermore, in the Supplementary Fig. 8, the local resolution maps are not sufficient for readers to evaluate the ligand density. Please show the ligands together with their surrounding residues simultaneously as isosurface or isomesh, at the same map threshold. At least two viewing angles should be provided.

We thank the reviewer for raising this important point. We have now included figures showing the ligands and surrounding residues simultaneously in mesh representation at the same threshold and in two different overall thresholds in Figures 2 e and f. Additional views are also shown in supplementary figures 8 d and e.

Reviewer #2 (Remarks to the Author):

The authors have altered the manuscript at several points, and have clearly added additional data, as requested. I would like to thank the authors for those efforts. Yet, I still see a lot of unclarities and confusing data. Overall, i remain at my point that the manuscript in its current state fails to provide conclusive support for its main conclusion. My major comments are listed below:

- The description of statistical methods remains suboptimal. In the methods section it is stated that "An unpaired nonparametric t-test followed by a Mann-Whitney U post-test" was used. I presume the Mann-Whitney test WAS the non-parametric t-test? It is also explicitly stated that no multi-group comparisons were performed. It seems highly unlikely that this is appropriate. For instance in Figs 3E, Suppl Fig 10A-C, Suppl Fig 13, a non-parametric ANOVA followed by a post-hoc test appears to me highly appropriate. Also, when EC50 and IC50 values are reported, no standard error or deviation is reported, and it is unclear how these values are compared to each other at several occasions in the MS.

Yes, it is correct; the statistical test used to compare the two groups was the unpaired Mann-Whitney test. As the reviewer suggested, this part has been modified to avoid misunderstanding.

Due to technical limitations, not all cells went through all the doses. Consequently, it was impossible for each cell to determine the IC50/EC50 and to perform a robust comparison between the different IC50s/EC50s determined. These IC50s/EC50s have been calculated using appropriate fit on the average dose-effect data for each variant, which also explains the absence of s.e.m or s.d. Due to this statistical limitation, we cannot compare the different IC50s/EC50s.

- The authors correctly cite the literature that an open pore TRPM4 structure is apparently difficult to obtain even in the Ca bound state. Nevertheless, my original point remains: it is completely unclear from the current set of data whether the binding site for anthranilic compounds which is identified from the CryoEM data is also available in the Ca bound state of the protein. Considering that in HsTRPM4 Ca is coordinated by Glu828, Gln831, Asn865, and Asp868 (PMID: 29217581) and that Ser863 is claimed to be an essential amino acid for NBA binding to TRPM4 (this work), it is very likely that Ca binding to the TRPM4 protein will alter the structure and availability of this binding site. The authors appear to address this by presenting structures of hsTRPM4 and mmTRPM4 in the presence of Ca, but these structures are derived in DDM/CHS. As the authors point out themselves, in DDM/CHS the binding site is occluded by a cholesterol molecule. The authors further appear to claim that microM Ca present in their SMA buffer would be sufficient to produce a Ca bound structure (Supp Fig 9a). However, this was never mentioned in the original structures (and thus, I presume, also not obvious), and also no supporting data for this claim is presented. None of these new structures (Ca bound HsTRPM4 and MmTRPM4) are decently described in the results section. How does the binding pocket for anthranilics look like in the Ca bound condition? Thus, my main point of criticism remains: the authors have identified critical amino acids for the interaction of anthranilic compounds to TRPM4, but I disagree that the available data unequivocally show that this binding pocket is also available in the Ca bound TRPM4 structure and thus, that the presence of anthranilic compounds in this binding pocket actually closes the channel. The possibility remains that these amino acids restrict the access of compounds to another binding site which only becomes accessible in the Ca bound state of the protein. In that respect the importance of the current findings is severely limited.

- The authors try to exclude that the binding site for anthranilic compounds and Ca interfere,

by presenting the Ca sensitivity of several TRPM4 variants and of TRPM4 in the presence of NBA. However, this assumption can certainly not be excluded from Suppl Fig 13b. How can one reliably determine the Ca sensitivity of a channel that is 90% blocked?

We appreciate the detailed feedback from the reviewer and apologise for the ambiguity in our previous explanation. Indeed, as well noted by the reviewer, in the three structures we determined of SMA extracted TRPM4, calcium ions were not included in our buffers; therefore, the sample was not supplemented with additional calcium. We agree that this is not the ideal experimental condition for structure determination. However, it is important to mention that the major technicality that enabled us to identify the inhibitor binding site was using SMA instead of detergents for extraction and purification. A drawback in the use of SMA is that SMA is destabilized by the addition of divalent ions and our attempts to purify TRPM4 in the presence of SMA and calcium resulted in strong precipitation of the sample (<https://doi.org/10.1042/BCJ20160723>). Therefore, structure determination in the presence of calcium using SMA is impossible. Nonetheless, the previous comments from the reviewer caused us to review our structure analysis more thoroughly, and indeed, we could observe clear density in the calcium-binding site. When a published structure of calcium-bound TRPM4 is superimposed into our maps, we observe a clear fitting of the calcium ion and its interacting residues into the density of SMA-extracted TRPM4. We also reproduced cryoEM maps of detergent-extracted TRPM4 in the presence of calcium and showed similar fitting. Although we cannot unequivocally say that it is a calcium ion and not another ion bound in our SMA-derived structures, at least this suggests that the residues involved in calcium binding are in a similar conformation to when in a calcium-bound state. Also, since EDTA is not added during our purification as done in previous studies to reveal its structure in the no calcium state (doi:[10.1126/science.aar4510](https://doi.org/10.1126/science.aar4510)), it is possible that our SMA-derived TRPM4 remains bound to endogenous calcium from the cell and still represents a calcium-bound state. To completely satisfy the reviewer's comments, we would require determining structures of TRPM4 directly from the cell; such studies are becoming more possible but remain very technically challenging and outside of the scope of our current study. Our study using SMA-extracted TRPM4 significantly advances the field, especially as a structure of TRPM4 bound to specific inhibitors has been elusive.

Furthermore, as the reviewer correctly stated, our previous hypothesis was that inhibitor binding destabilised the calcium-binding site due to the effects we observed with reduced TRPM4 currents in our mutants. However, upon the reviewer's prompting, we have completely revised this hypothesis as we did not observe any reduction in calcium sensitivity with our mutants or the presence of the inhibitor. We thank the reviewer for this.

These points are now better described in the text in **lines 130 – 140**, a newly included **figure 4** and **supplementary figure 9**.

In addition, as the reviewer noted, it is correct that the decrease of the current mediated by the NBA is more challenging to record and may lead to misinterpretation. However, we always take care to have a good signal-to-noise ratio to be confident in the 'TRPM4 current' record

in the presence of the NBA. Although the current decreased after the application of the NBA, the remaining current was still recordable and can be analyzed using the proper fit approach to determine the IC50.

We also decided not to normalize the dose-effects curves to determine the EC50 to avoid any misinterpretation from the reader (the NBA effect will not be more observable) and also because any normalization should assume, as highlighted by the reviewer, that a plateau phase is reached. Nevertheless, we have included normalized curves in Figure 4b, in which it was only used for the double mutant and in the presence of NBA compared to the wild type to highlight more clearly to the readers that the calcium sensitivity is not reduced. We thought this to be an important point to include the main figures as we had initially thought the mutants would reduce calcium sensitivity.

And in neither trace it is clear whether a plateau has been reached that allows to delineate the dose-response curve.

Thanks to the reviewer for raising this valid point. Although a high free calcium concentration was used (up to 5 mM), the absence of a well-delineated plateau phase in the dose effect was also one of our concerns during the analysis process. However, using an equation without any constraints to fit all the conditions showed a good correlation between the data points and the fit ($0.80 < R^2 < 0.95$), suggesting that the "data set" is enough to estimate the different IC50s and EC50s. Nevertheless, no statistical comparisons have been performed between the different IC50s and EC50s due to technical limitations previously mentioned (not all cells went through all the doses).

And it is also unclear how EC50 values are statistically compared?

No comparison has been made between the IC50s and EC50s because, due to technical limitations, not all the doses have been performed in one trial on the same cells. Overall, no IC50 or EC50 can be calculated per cell, which avoids comparing those IC50s/EC50s between the different conditions.

When comparing the WT trace from panel 13b with the traces of mutants in panel 13a, clear differences in Ca sensitivity appear obvious: for instance Y1057A, R1064S, and S863A/S924A have an EC50 value which is almost 50% lower compared to WT, indeed indicating that the suggested binding site of anthranilic compounds and Ca interacts?

Thanks to the reviewer for this comment. It is correct that, although no statistical tests have been performed on EC50s, the calcium sensitivity between the different variants and the WT seems different. The EC50s of the variants seem lower than the WT, suggesting that the sensitivity to the calcium of these variants is higher than the WT TRPM4 channel. In conclusion, the decrease of current observed with these variants (in the presence of 300 μ M calcium free) is probably not due to decreased calcium sensitivity. The reduction in the current of these variants may be due to a defect of expression, localization, or another alteration of the biophysical parameters. The evidence of the interconnection between the

NBA and calcium binding sites is complex to draw based on the data. The variation of the amino acids present in this study may lead to structural modifications of the closely located calcium-binding site without meaning that in physiological conditions, with the non-mutated TRPM4 channel, the interaction of NBA compound with TRPM4 protein interferes with the calcium-sensitivity of the channel by decreasing this latest. The fact that the variants investigated, for which the NBA sensitivity is reduced, seem more sensitive to calcium does not correlate with the model in which the decrease of the TRPM4 current mediated by NBA application is due to a reduction in calcium sensitivity.

Thus, I remain at my point that a CryoEM structure in the presence of Ca and anthranilics is essential for a proper conclusion whether the anthranilic binding site which is suggested here is also present in the Ca bound protein, or that other conclusive data should be presented that Ca binding to TRPM4 does not interfere with the anthranilic binding site.

- Why exactly the Ca dependence of the specific variants in Fig13 is determined is also not clear at all. When correlating data with SupplFig 12, Y1057A, Q1061A and R1064S show hardly any activity in the patches. Again, how can one reliably determine the Ca dependence of these loss-of-function variants.

Thanks to the reviewer for this comment. Concerning the cryoEM structure, as previously mentioned, we do see that the calcium-binding site is occupied with density in our structures of SMA-extracted TRPM4. We propose that endogenous calcium could be bound, but we cannot rule out that it is another ion. However, structure determination in the presence of calcium is a major limitation in using SMA. Our calcium sensitivity curves suggest these variants do not destabilize TRPM4 calcium sensitivity. Taken together, these results suggest that the calcium ion binding does not interfere with the anthranilic binding site. The variants used have been selected based on the substantial decrease, compared to the wild-type of the TRPM4 current using 300 μ M of free calcium. Previous comments from the reviewers suggested that the absence of current recorded at 300 μ M of free calcium may be due to a decrease in calcium sensitivity related to the mutation of the NBA binding pocket. The data reported using the three loss-of-function variants suggested that the reduction in current may not be due to a decrease in calcium sensitivity (the EC50s of these variants are lower than the wild-type EC50). We agree with the reviewer concerning the determination of those EC50s. However, as previously discussed, the good correlation between our data points and the fit used leads us to trust the values reported in this study.

On the other hand, S924A appears from Fig 12 a gain-of-function variant, but unfortunately Ca²⁺ dependence of this variant is not determined?

Thanks for this comment. More experiments have been performed using this variant before submitting this manuscript to investigate if this variant leads to a gain-of-function, as mentioned by the reviewer. Overall, no statistical differences were observed between the WT TRPM4 and the p.S924A TRPM4 variant (Itrpm4 WT = 1027 \pm 136 pA (n=15), Itrpm4 p.S924A = 1665 \pm 443 (n=11); p=0.4495). Because the current was not statistically different, we considered that there is no need to determine the calcium sensitivity (contrary to the loss-of-function).

- I don't understand the rationale behind the use of IBA, instead of CBA that was extensively characterized by the (co-)authors? Are properties of IBA and CBA the same? IBA blocks the channel (though that is not so clear from Fig 2B, see my point below), but is it also human specific, as CBA? If it binds in the same binding pocket as NBA, how do the authors then explain that the one is HsTRPM4 specific and the other is not, or is this not the case? This is a very important topic, which deserves more attention.

Thanks to the reviewer for this question. We opted to use IBA in this study because it had a similar IC₅₀ to NBA and although IBA has a similar structure to CBA, we presumed that the heavier iodine atom would give a relatively stronger signal for structure determination. It is correct that, based on the data, NBA, CBA, and IBA block wild-type human TRPM4 channels. Although the CBA cannot block mouse TRPM4 channels, no experiments have been performed to investigate the effect of IBA on mouse TRPM4 currents. It will be interesting to examine the impact of IBA on mouse TRPM4 to know if or not this compound, as CBA, is species-specific.

Interestingly, the putative mouse NBA binding pocket's amino acid composition differs from that of humans by only one amino acid. Experiments mutating this amino acid to mimic the 'NBA mouse binding pocket' in the human TRPM4 channel did not abolish the inhibitory effect of CBA on this chimera, suggesting that other molecular mechanisms are involved in this species' specificity of the compounds. Further experiments must be performed to investigate this question, which unfortunately cannot be part of this manuscript.

Furthermore, the authors assert that the interactions between the anthranilic acid moiety and residues of TRPM4 elucidate the chemical foundation underlying the specific binding of anthranilic acid derivatives to TRPM4. However, as illustrated in Figure 3, the orientation of the anthranilic moiety differs fundamentally, with the carboxyl of NBA facing S3 and S4, and of IBA facing the S4-S5 linker. The anthranilic moiety serves as a core structure for this class of TRPM4 antagonists, and is essential for their activity (PMID: 29579323). It is, at the very least, highly surprising that there exists such a significant difference in orientation, and this should be addressed by the authors.

-Coming back to Fig2B: I infer from the methods section that block of the channel throughout the paper is calculated at +100mV. Yet, at least for IBA (from Fig2B) it appears that at -100mV the effect is a lot smaller, or even absent? So IBA does not block inward currents? These data suggest that block of hsTRPM4 by IBA is voltage-dependent, is that the case? And is that also the case for NBA? From the traces it appears not, which would suggest that NBA and IBA have a completely different mode of action? Does it then make sense that they bind the same pocket? The authors should address this.

Thanks to the reviewer for this very important comment and for calling out the differences in the orientation of the molecules. We have reanalyzed the fitting of the ligands into the density and have better fitted them, and they are now more similar. We have changed this in **Figure 3**.

As presented, at +100 mV, the effects of IBA and NBA are similar. The reason for performing such analysis only at +100 mV is motivated by the high variability of current recorded at -100 mV in the HEK-293 cell line for an unknown reason. At -100 mV, the TRPM4 current (biophysical and amplitude) is surprisingly variable, and we are still not convinced that the quantification of this current at -100 mV is reliable. From our expertise in the HEK-293/TsA201 cell line, the TRPM4 current recorded at -100mV is so variable from

cell to cell in the presence or absence of a drug that we never quantify it, and we think that we must not draw any conclusion on this part.

Minor:

- It is somewhat disappointing that no whole cell currents of TRPM4 variants are reported. Supp Fig 10 suggests strong differences in activity (gain- and loss-of-function) of the different channel variants, but it is unclear whether this is the result of the variation in the number of channels in the excised patch. The variation in the western blot data is so large that hardly a conclusion is possible.

No whole-cell experiments have been conducted due to technical challenges, such as the requirement to reach the stability of the current (between 10-15 minutes in our experimental conditions) before applying any drugs. Moreover, the whole-cell approach is also imperfect due to other limitations (e.g., transfection efficiency issues from experiment to experiment or stability of the recording), which may also challenge the overall interpretation of the data.

In conclusion, we think that the complete analysis of these variants does not rely solely on whole-cell patch clamp experiments. Still, it must be conducted using different approaches, such as biotinylation experiments, immunostaining approaches, and single-channel recordings. Unfortunately, such an investigation is beyond the scope of this study.

- Overall, the text contains numerous typos and needs editing. Also, it appears to me that more data should be presented in the main figures. I suggest transferring data from the supplementary figures to the main figures.

We have reviewed the text and removed typos. Also, figures 3, 4, and 5 have been modified to include more data.

We thank the reviewer for the detailed comments, which we have now addressed, and we believe that they have significantly improved the clarity of our work. Point-to-point responses are detailed below in red:

Reviewer #2 (Remarks to the Author):

The authors have not fundamentally addressed the issues that I brought up in my previous review. It is clear that this is an extensive piece of work, but the issue of an interrelation between the IBA/NBA and the Ca binding site of the protein and the way that it is being treated in this MS remains an important issue in my view that limits the decisiveness of this work. This should at least be critically discussed in the discussion section of the paper.

Thanks for this comment. We have now included a discussion on this limitation of our study in the discussion section, lines 255-264 of the manuscript.

I specifically disagree with the statistical evaluation of EC50 and IC50 values, which is apparently impossible? What is against determining the EC50 value for each cell, and comparing them between WT and mutants? I advise that a statistical expert goes through the methodology of the paper.

Thanks for this comment, and we are sorry if our former answer was unclear.

I: For dose-effect curves related to the compounds (NBA)

To perform those dose-effect curves (Figure 5a in the new version of the manuscript), we used eight different NBA concentrations (10 nM, 50 nM, 100 nM, 500 nM, 1 μ M, 5 μ M, 10 μ M, and 50 μ M). However, our perfusion system has only seven channels, two of which need to be used for background and saturation measurements: We first need to apply a solution containing a concentration of 0 μ M of free calcium (without any compounds) to record the background current that is present without activation of the HsTRPM4 channel. We then activate the HsTRPM4 channel to record the calcium-activated sodium current using a solution containing a concentration of 300 μ M of free calcium (also without any compounds) (see material and methods of the new manuscript). After this activation and the stabilization of the calcium-activated sodium current from the HsTRPM4 channel, we could apply a solution containing a concentration of 300 μ M of free calcium (to keep the HsTRPM4 channel open/active) with a desired compound concentration.

This leaves only a maximum of five different concentrations of the compounds that can be applied per cell, as presented below:

Protocol of perfusion used to apply the five lower concentrations of the compounds on cells expressing HsTRPM4 channels.

Two perfusions are used to activate the HsTRPM4 channel (wildtype or variants) and quantify the calcium-activated sodium current.

- Perfusion channel n°1: Solution with 0 μ M of free calcium without NBA.
- Perfusion channel n°2: Solution with 300 μ M of free calcium without NBA.

Five perfusions are still available to investigate the effect of the NBA compounds on the calcium-activated sodium current from HsTRPM4 channels (wildtype or variants).

With the five lower compound doses.

- Perfusion channel n°3: Solution with 300 μ M of free calcium with 10 nM of NBA.

- Perfusion channel n°4: Solution with 300 μ M of free calcium with 50 nM of NBA.
- Perfusion channel n°5: Solution with 300 μ M of free calcium with 100 nM of NBA.
- Perfusion channel n°6: Solution with 300 μ M of free calcium with 500 nM of NBA.
- Perfusion channel n°7: Solution with 300 μ M of free calcium with 1 μ M of NBA.

For the 3 higher compound doses, applied to another cell than the one used for the 5 lower compound doses, the protocol is:

Protocol of perfusion used to apply the three higher concentrations of the compounds on cells expressing HsTRPM4 channels.

Two perfusions are used to activate the HsTRPM4 channel (wildtype or variants) and quantify the calcium-activated sodium current.

- Perfusion channel n°1: Solution with 0 μ M of free calcium without NBA.
- Perfusion channel n°2: Solution with 300 μ M of free calcium without NBA.

Five perfusions are still available to investigate the effect of the NBA compounds on the calcium-activated sodium current from HsTRPM4 channels (wildtype or variants). With the three higher compound doses.

- Perfusion channel n°3: Solution with 300 μ M of free calcium with 5 μ M of NBA.
- Perfusion channel n°4: Solution with 300 μ M of free calcium with 10 μ M of NBA.
- Perfusion channel n°5: Solution with 300 μ M of free calcium with 50 μ M of NBA.

Due to this technical limitation, we decided first to apply the 5 lower compound doses to different cells and then the 3 higher compound doses to cells distinct from the ones used for the lower compound doses.

Unfortunately, this approach does not allow us to calculate the IC₅₀ per cell or perform a statistical analysis between two groups.

Nevertheless, as the reviewer requested, we estimated the IC₅₀ for each cell per group by:

1. Taking only the doses applied to the same cell (the first five lower concentrations), and
2. Assuming that for a very high concentration of the compounds (1 mM), the percentage of inhibition of the calcium-activated sodium current from the HsTRPM4 channels (wildtype and variants) will be 100%.

Using this approach, the different IC_{50s} have been estimated as reported in the Table R1 below:

Mutation	n	Mean IC₅₀ (nM)	Std. error of the mean (nM)	Significance (WT vs. variant)
WT	6	130	10	-
S863A	6	926	85	$p < 0.05$
V901W	7	334	43	$p < 0.05$
H908A	6	820	41	$p < 0.05$
S924A	6	780	39	$p < 0.05$
R1064A	6	1164	222	$p < 0.05$
R1064G	6	1097	220	$p < 0.05$
S863A/S924A	6	7747	2015	$p < 0.05$
T667I	6	212	17	$p < 0.05$

Table R1: NBA inhibition IC₅₀ values for HsTRPM4 channels with specific mutations.

Following that estimation, an unpaired nonparametric t-test and a Mann-Whitney U post-test were used to compare two unpaired groups. A p-value of $p < 0.05$ was considered significant.

Comparing the WT condition to all the others showed a significant difference (WT vs. variant) (Table R1, significance column).

However, based on the approach of fixing the maximum percentage of inhibition for an arbitrary compound concentration, we do not want to present these data in the manuscript, as this approach violates the rule of independence, a fundamental rule for statistics, as also confirmed by the statisticians contacted (Dr. Romain-Daniel Gosselin).

The option proposed by the statistician is to perform a two-way ANOVA with two-factor repeated measures to investigate if there is a difference between two groups (WT vs. variant) for a given concentration of the compounds (NBA) from the five first lower compounds concentrations applied on the same cell.

To perform those comparisons, the concentration of 1 μ M of a compound (NBA) has been chosen. We decided to take this concentration of the compound of NBA (1 μ M) because this concentration is relatively close to the IC_{50s} of many HsTRPM4 channel variants investigated in this manuscript (see Table R1 above, Mean IC₅₀).

As shown below in Table R2, except for the HsTRPM4 T677I mutant, the percentages of inhibition for all variants at the concentration of 1 μ M of compound significantly differed ($p < 0.05$) from the WT and are smaller, suggesting a decrease of inhibition mediated by the compounds, on these variants of HsTRPM4 channel.

Mutation	n	Percentage of inhibition for 1 μ M of NBA (%)	Std. error of the mean (%)	Significance (WT vs. variant)
WT	6	96.5	1.5	-
S863A	6	58.6	4.3	$p < 0.05$
V901W	7	78.6	1.4	$p < 0.05$
H908A	6	57.1	2.1	$p < 0.05$
S924A	6	68.4	3.8	$p < 0.05$
R1064A	6	51.5	5.2	$p < 0.05$
R1064G	6	56.8	7.0	$p < 0.05$
S863A/S924A	6	31.4	2.9	$p < 0.05$
T667I	6	93.2	1.5	ns.

Table R2: Percentage of inhibition of the calcium-activated sodium current at a concentration of 1 μ M of NBA for HsTRPM4 channels with specific mutations.

To clarify the analysis, we have explained this limitation from the limited number of perfusion channels for the IC₅₀ determination in the manuscript now, and we also included the alternative approach proposed by the statistician to the data analysis and statistic section, lines 890-908 of the manuscript.

II: For dose-effect curves related to the calcium concentration

II.A: For the dose-effect curves related to the calcium concentration without any vehicle or NBA

For measuring the calcium sensitivity curves (Supplementary Figure 13a in the new version of the manuscript), all seven calcium concentrations were applied to the same cell. This is possible, because the values of 0 μ M and the 300 μ M are part of the concentrations of free calcium analyzed in this experiment.

In this experiment, we do not have any compounds (NBA), meaning that we can use the solutions containing 0 μ M and 300 μ M of free calcium to evaluate the effect of calcium on

the calcium-activated sodium current of HsTRPM4. In doing so, five other solutions containing different concentrations of free calcium can be measured, as presented below:

Two perfusions are used to activate the HsTRPM4 channel (wildtype or variants) and quantify the calcium-activated sodium current.

- Perfusion channel n°1: Solution with 0 μM of free calcium without NBA.
- Perfusion channel n°2: Solution with 300 μM of free calcium without NBA.

Five perfusions are used to activate the HsTRPM4 channel (wildtype or variants) and quantify the calcium-activated sodium current.

- Perfusion channel n°3: Solution with 50 μM of free calcium without NBA.
- Perfusion channel n°4: Solution with 100 μM of free calcium without NBA.
- Perfusion channel n°5: Solution with 500 μM of free calcium without NBA.
- Perfusion channel n°6: Solution with 1000 μM of free calcium without NBA.
- Perfusion channel n°7: Solution with 5000 μM of free calcium without NBA.

Based on this approach, an EC_{50} for each cell can be calculated. Then, an unpaired nonparametric t-test and a Mann-Whitney U post-test were used to compare the EC_{50} s from the different conditions. As mentioned in the new manuscript in line 190-192: "the different EC_{50} s of the variants are smaller than the wildtype EC_{50} , suggesting an increase in calcium sensitivity." Overall, these data do not support the assumption that the different amino acid alterations performed in the drug binding pocket leading to a loss of function of the HsTRPM4 channel and a decrease of the efficiency of NBA are due to a decrease of the calcium sensitivity of those HsTRPM4 variant channels

II.B: For the dose-effect curves related to the calcium concentration with NBA

Concerning the calcium sensitivity curve with NBA (Figure 4b and Supplementary Figure 13b in the new version of the manuscript), all seven concentrations have been applied to the same cell, because in this experiment, all solutions contain 200 nM NBA, including the 0 μM and 300 μM free calcium concentrations. In doing so, five other solutions containing different concentrations of free calcium with 200 nM NBA were tested, as presented below:

Two perfusions are used to activate the HsTRPM4 channel (wildtype or variants) and quantify the calcium-activated sodium current.

- Perfusion channel n°1: Solution with 0 μM free calcium and 200 nM of NBA.
- Perfusion channel n°2: Solution with 300 μM of free calcium and 200 nM of NBA.

Five perfusions are used to activate the HsTRPM4 channel (wildtype or variants) and quantify the calcium-activated sodium current.

- Perfusion channel n°3: Solution with 50 μM of free calcium and 200 nM of NBA.
- Perfusion channel n°4: Solution with 100 μM of free calcium and 200 nM of NBA.
- Perfusion channel n°5: Solution with 500 μM of free calcium and 200 nM of NBA.
- Perfusion channel n°6: Solution with 1000 μM of free calcium and 200 nM of NBA.
- Perfusion channel n°7: Solution with 5000 μM of free calcium and 200 nM of NBA.

Based on this approach, an EC_{50} for each cell was calculated. Then, an unpaired nonparametric t-test and a Mann-Whitney U post-test were used to compare the EC_{50} s from the different conditions. As mentioned in the new manuscript in lines 195-197: "As for the variants, the results suggest that NBA does not decrease the calcium sensitivity of the wildtype channels."

Furthermore, about my comment on the orientation of NBA and IBA in the binding pocket of

the original MS, the authors now comment that "We have reanalyzed the fitting of the ligands into the density and have better fitted thm (sic), and they are now more similar. We have changed this in Figure 3." I'm very surprised by the looseness of this adaptation. Was the previous fitting not well performed? Which parameters determine whether the compounds are "better fitted"? To me this indicates that there is considerable freedom of the compounds in this binding pocket and that it is unclear how the authors can determine the "correct" orientation of IBA or NBA in this binding pocket. This uncertainty should be highlighted in the manuscript, and is worth to be discussed.

To further clarify our previous comment. Each TRPM4 molecule has 4 ligand (IBA) binding sites. Therefore, we have to fit the ligands (NBA/IBA) 4 times into the structure. In the previous version of Figure 3 we had properly fitted the ligand in three of the binding sites, but not the fourth one, and unfortunately, we showed the wrongly fitted one in the figure. We recognized this only later. This has now been fixed. More information about placement of the ligand is found in lines 147-152. In any case, the 3D maps and the fits are available at the EMDB and PDB databases, respectively, as detailed in the Data and Code Availability section in lines 921-925.

I also find the treatment of TRPM4 currents at -100mV as unreliable ("...we never quantify it, and we think that we must not draw any conclusion on this part") quite disturbing. Many papers have analysed in the past both inward and outward currents through TRPM4. Negative potentials are off course the physiological range in which the channel is active, and the fact that currents in the presence of the blocker are very variable in this voltage range might well indicate that the block of TRPM4 by anthranilic compounds is voltage dependent. At least, this way of analysing and the reason for it should be clearly highlighted in the manuscript, so that it is clear to the reader why this is done in this way.

As the reviewer noted, we also noticed that the inhibition efficiency mediated by the treatment (NBA) on a cell expressing the HsTRPM4 channel sometimes differs at -100 mV compared to the one observed at + 100 mV. However, this effect at -100 mV is challenging to quantify, mainly because this current is small after the stabilization of the calcium-activated sodium current from the HsTRPM4 channel. As shown in the voltage-current relationship presented here below (Figure n°1), in our conditions, the calcium-activated sodium current from the HsTRPM4 (Human TRPM4) channel recorded at -100 mV (after stabilization) is smaller than the one recorded in a similar condition with the MsTRPM4 (Mouse TRPM4) for biological reasons that we cannot explain.

Figure n°1: Normalized current-voltage relationship curves for human and mouse TRPM4 isoforms recorded after the stabilization of the calcium-activated sodium currents.

Due to this phenomenon, quantifying the effect of the treatment (NBA) at -100 mV using the human isoform can lead to a misestimation of the effects due to the small currents at negative voltages, as shown in Figure n°2 below.

A

B

Figure n°2: Calcium-activated sodium currents from HsTRPM4 (human TRPM4) channel in the absence of a compound (black traces) and after perfusion of the compounds (blue traces panel A for NBA or panel B for CBA). The small amplitude of the calcium-activated sodium currents from HsTRPM4 at -100 mV without any treatment (black traces) renders the quantification of the drug effect at this voltage during the last 100 ms of the pulse (green area) challenging compared to the quantification performed at + 100 mV during the last 100 ms of the pulse (yellow area).

Consequently, to avoid misinterpreting the data, we did not quantify the effect of the compounds at -100 mV during the last 100 ms. Overall, for the time being, we are unable to draw any conclusion concerning the voltage dependence of the block on calcium-activated sodium currents from HsTRPM4 mediated by anthranilic compounds.

The information concerning the non-analysis of the treatment effect at -100 mV has now been added and explained in the new version of the manuscript in the legend of the supplementary figure 11 and the methods section lines 872-879.

Finally, during the review process an important manuscript in the same field was published (Hu et al., Nature, PMID: 38750366). They highlight the importance of temperature for the confirmation of the TRPM4 protein. The implications of this work for the current manuscript should be addressed in the discussion.

We have addressed this in the discussion section of the manuscript from lines 239-248.

We thank the reviewer for the detailed comments, which we have now addressed, and we believe that they have significantly improved the clarity of our work. Point-to-point responses are detailed below in red:

Reviewer #2 (Remarks to the Author):

It should be more clear from the legend and make-up of Figure 4a and Suppl Fig 9 (top panel) that the authors cannot exclude that the density observed in the Ca binding site is not actually a Ca ion (as correctly outlined in the results section, lines 130-144). Now it is clearly depicted as a Ca ion (green ball, with Ca annotated). I suggest another color (compared to the DDM:CHS condition where more confidence exists that Ca) and removing Ca as annotation in the figure. Actually, in the method section on protein solubilisation there is no mention of how much Ca was added in the DDM:CHS condition, nor is it obvious how much 'endogenous Ca' could be expected in the SMA condition. One could assume that in the absence of any Ca buffer and using purified components that lower microM concentrations of Ca are still present in the 'no Ca added condition', but as is obvious to the authors, this is hardly sufficient to activate TRPM4, and thus it is actually highly unlikely that Ca is present in the Ca binding pocket in the SMA-no added Ca condition. I'm satisfied with how this is handled in the text, but the figure legend and the figure itself should reflect this interpretation.

We thank the reviewer for this comment. We have removed all the Ca ion annotations in the figures and figure legends as requested.